# Sequential Test for the Lowest Mean: From Thompson to Murphy Sampling

**Emilie Kaufmann**[1]    **Wouter M. Koolen**[2]    **Aurélien Garivier**[3]

[1] CNRS & U. Lille, CRIStAL / SequeL Inria Lille, `emilie.kaufmann@univ-lille.fr`
[2] Centrum Wiskunde & Informatica, Amsterdam, `wmkoolen@cwi.nl`
[3] UMPA, École normale supérieure de Lyon, `aurelien.garivier@ens-lyon.fr`

## Abstract

Learning the minimum/maximum mean among a finite set of distributions is a fundamental sub-task in planning, game tree search and reinforcement learning. We formalize this learning task as the problem of sequentially testing how the minimum mean among a finite set of distributions compares to a given threshold. We develop refined non-asymptotic lower bounds, which show that optimality mandates very different sampling behavior for a low vs high true minimum. We show that Thompson Sampling and the intuitive Lower Confidence Bounds policy each nail only one of these cases. We develop a novel approach that we call Murphy Sampling. Even though it entertains exclusively low true minima, we prove that MS is optimal for both possibilities. We then design advanced self-normalized deviation inequalities, fueling more aggressive stopping rules. We complement our theoretical guarantees by experiments showing that MS works best in practice.

## 1 Introduction

We consider a collection of core problems related to *minimums of means*. For a given finite collection of probability distributions parameterized by their means $\mu_1, \ldots, \mu_K$, we are interested in learning about $\mu^* = \min_a \mu_a$ from adaptive samples $X_t \sim \mu_{A_t}$, where $A_t$ indicates the distribution sampled at time $t$. We shall refer to these distributions as arms in reference to a multi-armed bandit model [28, 26]. Knowing about minima/maxima is crucial in reinforcement learning or game-playing, where the value of a state for an agent is the *maximum* over actions of the (expected) successor state value or the *minimum* over adversary moves of the next state value.

The problem of estimating $\mu^* = \min_a \mu_a$ was studied in [34] and subsequently [7, 31, 8]. It is known that no unbiased estimator exists for $\mu^*$, and that estimators face an intricate bias-variance trade-off. Beyond estimation, the problem of constructing *confidence intervals* on minima/maxima naturally arises in (Monte Carlo) planning in Markov Decision Processes [15] and games [25]. Such confidence intervals are used hierarchically for Monte Carlo Tree Search (MCTS) in [32, 11, 17, 20]. The open problem of designing asymptotically optimal algorithms for MCTS led us to isolate one core difficulty that we study here, namely the construction of confidence intervals and associated sampling/stopping rules for learning minima (and, by symmetry, maxima).

Confidence intervals (that are uniform over time) can be naturally obtained from a (sequential) test of $\{\mu^* < \gamma\}$ versus $\{\mu^* > \gamma\}$, given a threshold $\gamma$. The main focus of the paper goes even further and investigates the minimum number of samples required for *adaptively* testing whether $\{\mu^* < \gamma\}$ or $\{\mu^* > \gamma\}$, that is sequentially sampling the arms in order to decide for one hypothesis as quickly as possible. Such a problem is interesting in its own right as it naturally arises in several statistical certification applications. As an example we may consider quality control testing in manufacturing, where we want to certify that in a batch of machines each has a guaranteed probability of successfully producing a widget. In e-learning, we may want to certify that a given student has

sufficient understanding of a range of subjects, asking as few questions as possible about the different subjects. Then in anomaly detection, we may want to flag the presence of an anomaly faster the more anomalies are present. Finally, in a crowdsourcing system, we may need to establish as quickly as possible whether a cohort of workers contains at least one unacceptably careless worker. Our own motivation for studying this problem is that it corresponds to an especially simple instance of the depth-two game tree search problem, as illustrated in Figure 1.

We thus study a particular example of sequential adaptive hypothesis testing problem, as introduced by Chernoff [5], in which multiple experiments (sampling from one arm) are available to the experimenter, each of which allows to gain different information about the hypotheses. The experimenter sequentially selects which experiment to perform, when to stop and then which hypothesis to recommend. Several recent works from the bandit literature fit into this framework, with the twist that they consider continuous, composite hypotheses and aim for $\delta$-correct testing: the probability of guessing a wrong hypothesis has to be smaller than $\delta$, while performing as few experiments as possible. The fixed-confidence *Best Arm Identification* problem (concerned with finding the arm with largest mean) is one such example [9, 23], of which several variants have been studied [19, 17, 12]. For example the Thresholding Bandit Problem [27] aims at finding the set of arms above a threshold, which is strictly harder than our testing problem. In the Ranking

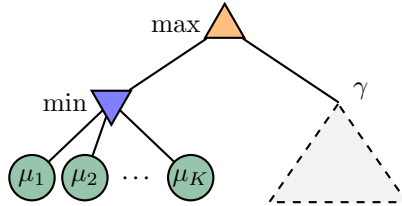

Figure 1: Game tree search problem of "depth $1^1/2$". We consider the scenario where it has been established that the right subtree (grey) of the root has value $\gamma$. Learning the optimal action at the root (orange) is equivalent to determining how the minimum (blue) of the leaf means (green) compares to $\gamma$.

and Selection literature (see e.g., [14] for a survey) the related problem of finding systems whose expected performance is smaller than a known standard has been studied by [24], but if such system exist, the goal was to additionnaly identify the one with smallest expectation, which is strictly harder than our problem.

A full characterization of the asymptotic complexity of the BAI problem was recently given in [11], highlighting the existence of an *optimal allocation of samples* across arms. The lower bound technique introduced therein can be generalized to virtually any testing problem in a bandit model (see, e.g. [20, 12]). Such an optimal allocation is also presented by [4] in the GENERAL-SAMP framework, which is quite generic and in particular encompasses testing on which side of $\gamma$ the minimum falls. The proposed LPSample algorithm is thus a candidate to be applied to our testing problem. However, this algorithm is only proved to be *order-optimal*, that is to attain the minimal sample complexity up to a (large) multiplicative constant. Moreover, like other algorithms for special cases (e.g. Track-and-Stop for BAI [11]), it relies on *forced exploration*, which may be harmful in practice and leads to unavoidably asymptotic analysis.

Our first contribution is a tight lower bound on the sample complexity that provides an oracle sample allocation, but also aims at reflecting the moderate-risk behavior of a $\delta$-correct algorithm. Our second contribution is a new sampling rule for the minimum testing problem, under which the empirical fraction of selections converges to the optimal allocation without forced exploration. The algorithm is a variant of Thompson Sampling [33, 1] that is conditioning on the "worst" outcome $\mu^* < \gamma$, hence the name Murphy Sampling. This conditioning is inspired by the Top Two Thompson Sampling recently proposed by [29] for Best Arm Identification. As we shall see, the optimal allocation is very different whether $\mu^* < \gamma$ or $\mu^* > \gamma$ and yet Murphy Sampling automatically adopts the right behavior in each case. Our third contribution is a new stopping rule, that by aggregating samples from several arms that look small may lead to early stopping whenever $\mu^* < \gamma$. This stopping rule is based on a new self-normalized deviation inequality for exponential families (Theorem 7) of independent interest. It generalizes results obtained by [18, 23] in the Gaussian case and by [3] without the uniformity in time, and also handles subsets of arms.

The rest of the paper is structured as follows. In Section 2 we introduce our notation and formally define our objective. In Section 3, we present lower bounds on the sample complexity of sequential tests for minima. In particular, we compute the optimal allocations for this problem and discuss the limitation of naive benchmarks to attain them. In Section 4 we introduce Murphy sampling, and prove its optimality in conjunction with a simple stopping rule. Improved stopping rules (and

confidence intervals) are presented in Section 5. Finally, numerical experiments reported in Section 6 demonstrate the efficiency of Murphy Sampling paired with our new stopping rule.

## 2 Setup

We consider a family of $K$ probability distributions that belong to a one-parameter canonical exponential family, that we shall call *arms* in reference to a multi-armed bandit model. Such exponential families include Gaussian with known variance, Bernoulli, Poisson, see [3] for details. For natural parameter $\nu$, the density of the distribution w.r.t. carrier measure $\rho$ on $\mathbb{R}$ is given by $e^{x\nu - b(\nu)}\rho(\mathrm{d}x)$, where the cumulant generating function $b(\nu) = \ln\mathbb{E}_\rho[e^{X\nu}]$ induces a bijection $\nu \mapsto \dot{b}(\nu)$ to the mean parameterization. We write $\mathrm{KL}(\nu, \lambda)$ and $d(\mu, \theta)$ for the Kullback-Leibler divergence from natural parameters $\nu$ to $\lambda$ and from mean parameters $\mu$ to $\theta$. Specifically, with convex conjugate $b_*$,

$$\mathrm{KL}(\nu, \lambda) = b(\lambda) - b(\nu) + (\nu - \lambda)\dot{b}(\nu) \quad \text{and} \quad d(\mu, \theta) = b_*(\mu) - b_*(\theta) - (\mu - \theta)\dot{b}_*(\theta).$$

We denote by $\boldsymbol{\mu} = (\mu_1, \dots, \mu_K) \in \mathcal{I}^K$ the vector of arm means, which fully characterizes the model. In this paper, we are interested in the smallest mean (and the arm where it is attained)

$$\mu^* = \min_a \mu_a \qquad \text{and} \qquad a^* = a^*(\boldsymbol{\mu}) = \arg\min_a \mu_a.$$

Given a threshold $\gamma \in \mathcal{I}$, our goal is to decide whether $\mu^* < \gamma$ or $\mu^* > \gamma$. We introduce the hypotheses

$$\mathcal{H}_< = \{\boldsymbol{\mu} \in \mathcal{I}^K \mid \mu^* < \gamma\} \quad \text{and} \quad \mathcal{H}_> = \{\boldsymbol{\mu} \in \mathcal{I}^K \mid \mu^* > \gamma\}, \quad \text{and their union} \quad \mathcal{H} = \mathcal{H}_< \cup \mathcal{H}_>.$$

We want to propose a sequential and adaptive testing procedure, that consists in a *sampling rule* $A_t$, a *stopping rule* $\tau$ and a *decision rule* $\hat{m} \in \{<, >\}$. The algorithm samples $X_t \sim \mu_{A_t}$ while $t \leq \tau$, and then outputs a decision $\hat{m}$. We denote the information available after $t$ rounds by $\mathcal{F}_t = \sigma(A_1, X_1, \dots, A_t, X_t)$. $A_t$ is measurable with respect to $\mathcal{F}_{t-1}$ an possibly some exogenous random variable, $\tau$ is a stopping time with respect to this filtration and $\hat{m}$ is $\mathcal{F}_\tau$-measurable.

Given a risk parameter $\delta \in (0, 1]$, we aim for a $\delta$-*correct* algorithm, that satisfies $\mathbb{P}_{\boldsymbol{\mu}}(\boldsymbol{\mu} \in \mathcal{H}_{\hat{m}}) \geq 1 - \delta$ for all $\boldsymbol{\mu} \in \mathcal{H}$. Our goal is to build $\delta$-correct algorithms that use a small number of samples $\tau_\delta$ in order to reach a decision. In particular, we want the *sample complexity* $\mathbb{E}_{\boldsymbol{\mu}}[\tau]$ to be small.

**Notation** We let $N_a(t) = \sum_{s=1}^t \mathbb{1}_{(A_s = a)}$ be the number of selections of arm $a$ up to round $t$, $S_a(t) = \sum_{s=1}^t X_s \mathbb{1}_{(A_s = a)}$ be the sum of the gathered observations from that arm and $\hat{\mu}_a(t) = S_a(t)/N_a(t)$ their empirical mean.

## 3 Lower Bounds

In this section we study information-theoretic sample complexity lower bounds, in particular to find out what the problem tells us about the behavior of oracle algorithms. [10] prove that for any $\delta$-correct algorithm

$$\mathbb{E}_{\boldsymbol{\mu}}[\tau] \geq T^*(\boldsymbol{\mu})\mathrm{kl}(\delta, 1 - \delta) \qquad \text{where} \qquad \frac{1}{T^*(\boldsymbol{\mu})} = \max_{\boldsymbol{w} \in \triangle} \min_{\boldsymbol{\lambda} \in \mathrm{Alt}(\boldsymbol{\mu})} \sum_a w_a d(\mu_a, \lambda_a) \quad (1)$$

$\mathrm{kl}(x, y) = x\ln\frac{x}{y} + (1 - x)\ln\frac{1-x}{1-y}$ and $\mathrm{Alt}(\boldsymbol{\mu})$ is the set of bandit models where the correct recommendation differs from that on $\boldsymbol{\mu}$. The following result specialises the above to the case of testing $\mathcal{H}_<$ vs $\mathcal{H}_>$, and gives explicit expressions for the *characteristic time* $T^*(\boldsymbol{\mu})$ and *oracle weights* $\boldsymbol{w}^*(\boldsymbol{\mu})$.

**Lemma 1.** *Any $\delta$-correct strategy satisfies* (1) *with*

$$T^*(\boldsymbol{\mu}) = \begin{cases} \frac{1}{d(\mu^*, \gamma)} & \mu^* < \gamma, \\ \sum_a \frac{1}{d(\mu_a, \gamma)} & \mu^* > \gamma, \end{cases} \qquad and \qquad w_a^*(\boldsymbol{\mu}) = \begin{cases} \mathbb{1}_{a = a^*} & \mu^* < \gamma, \\ \frac{\frac{1}{d(\mu_a, \gamma)}}{\sum_j \frac{1}{d(\mu_j, \gamma)}} & \mu^* > \gamma. \end{cases}$$

Lemma 1 is proved in Appendix B. As explained by [10] the oracle weights correspond to the fraction of samples that should be allocated to each arm under a strategy matching the lower bound. The interesting feature here is that the lower bound indicates that an oracle algorithm should have very different behavior on $\mathcal{H}_<$ and $\mathcal{H}_>$. On $\mathcal{H}_<$ it should sample $a^*$ (or all lowest means, if there are several) exclusively, while on $\mathcal{H}_>$ it should sample *all* arms with certain specific proportions.

## 3.1 Boosting the Lower Bounds

Following [13] (see also [30] and references therein), Lemma 1 can be improved under very mild assumptions on the strategies. We call a test *symmetric* if its sampling and stopping rules are invariant by conjugation under the action of the group of permutations on the arms. In that case, if all the arms are equal, then their expected numbers of draws are equal. For simplicity we assume $\mu_1 \leq \ldots \leq \mu_K$.

**Proposition 2.** *Let $k = \max_a d(\mu_a, \gamma) = \max\{d(\mu_1, \gamma), d(\mu_K, \gamma)\}$. For any symmetric, $\delta$-correct test, for all arms $a \in \{1, \ldots, K\}$, the expected number of selections of arm $a$ satisfies*

$$\mathbb{E}_{\boldsymbol{\mu}}[N_a(\tau)] \geq \frac{2\left(1 - 2\delta K^3\right)}{27K^2 k} .$$

Proposition 2 is proved in Appendix B. It is an open question to improve the dependency in $K$ in this bound; moreover, one may expect a bound decreasing with $\delta$, maybe in $\ln(\ln(1/\delta))$ (but certainly not in $\ln(1/\delta)$). This result already has two important consequences: first, it shows that even an optimal algorithm needs to draw all the arms a certain number of times, even on $\mathcal{H}_<$ where Lemma 1 may suggest otherwise. Second, this lower bound on the number of draws of each arm can be used to "boost" the lower bound on $\mathbb{E}_{\boldsymbol{\mu}}[\tau]$: the following result is also proved in Appendix B.

**Theorem 3.** *When $\mu^* < \gamma$, for any symmetric, $\delta$-correct strategy,*

$$\mathbb{E}_{\boldsymbol{\mu}}[\tau] \geq \frac{\mathrm{kl}(\delta, 1-\delta)}{d(\mu_1, \gamma)} + \frac{2\left(1 - 2\delta K^3\right)}{27K^2 k} \sum_a \left(1 - \frac{d(\mu_a, \gamma)\mathbb{1}_{(\mu_a \leq \gamma)}}{d(\mu_1, \gamma)}\right) .$$

## 3.2 Lower Bound Inspired Matching Algorithms

In light of the lower bound in Lemma 1, we now investigate the design of optimal learning algorithms (sampling rule $A_t$ and stopping rule $\tau$). We start with the stopping rule. The first stopping rule that comes to mind consists in comparing *separately* each arm to the threshold and stopping when either one arm looks significantly below the threshold or all arms look significantly above. Introducing $d^+(u, v) = d(u, v)\mathbb{1}_{(u \leq v)}$ and $d^-(u, v) = d(u, v)\mathbb{1}_{(u \geq v)}$, we let

$$\tau_{\mathrm{Box}} = \tau_< \wedge \tau_> \qquad \text{where} \qquad \begin{aligned} \tau_< &= \inf\left\{t \in \mathbb{N}^* : \exists a\, N_a(t)d^+(\hat{\mu}_a(t), \gamma) \geq C_<(\delta, N_a(t))\right\}, \\ \tau_> &= \inf\left\{t \in \mathbb{N}^* : \forall a\, N_a(t)d^-(\hat{\mu}_a(t), \gamma) \geq C_>(\delta, N_a(t))\right\}, \end{aligned} \qquad (2)$$

and $C_<(\delta, r)$ and $C_>(\delta, r)$ are two *threshold functions* to be specified. Box refers to the fact that the decision to stop relies on individual "box" confidence intervals for each arm, whose endpoints are

$$\begin{aligned} \mathrm{U}_a(t) &= \max\{q : N_a(t)d^+(\hat{\mu}_a(t), q) \geq C_<(\delta, N_a(t))\}, \\ \mathrm{L}_a(t) &= \min\{q : N_a(t)d^-(\hat{\mu}_a(t), q) \geq C_>(\delta, N_a(t))\}. \end{aligned}$$

Indeed, $\tau_{\mathrm{Box}} = \inf\{t \in \mathbb{N}^* : \min_a \mathrm{U}_a(t) \leq \gamma \text{ or } \min_a \mathrm{L}_a(t) \geq \gamma\}$. In particular, if $\forall a, \forall t \in \mathbb{N}^*, \mu_a \in [\mathrm{L}_a(t), \mathrm{U}_a(t)]$, any algorithm that stops using $\tau_{\mathrm{Box}}$ is guaranteed to output a correct decision. In the Gaussian case, existing work [18, 23] permits to exhibit thresholds of the form $C_{\lessgtr}(\delta, r) = \ln(1/\delta) + a \ln\ln(1/\delta) + b \ln(1 + \ln(r))$ for which this sufficient correctness condition is satisfied with probability larger than $1 - \delta$. Theorem 7 below generalizes this to exponential families.

Given that $\tau_{\mathrm{Box}}$ can be proved to be $\delta$-correct *whatever the sampling rule*, the next step is to propose sampling rules that, coupled with $\tau_{\mathrm{Box}}$, would attain the lower bound presented in Section 3. We now show that a simple algorithm, called LCB, can do that for all $\boldsymbol{\mu} \in \mathcal{H}_>$. LCB selects at each round the arm with smallest Lower Confidence Bound:

$$\boxed{\texttt{LCB: Play } A_t = \mathrm{argmin}_a \mathrm{L}_a(t) ,} \qquad (3)$$

which is intuitively designed to attain the stopping condition $\min_a \mathrm{L}_a(t) \geq \gamma$ faster. In Appendix E we prove (Proposition 15) that LCB is optimal for $\boldsymbol{\mu} \in \mathcal{H}_>$ however we show (Proposition 16) that on instances of $\mathcal{H}_<$ it draws all arms $a \neq a^*$ too much and cannot match our lower bound.

For $\boldsymbol{\mu} \in \mathcal{H}_<$, the lower bound Lemma 1 can actually be a good guideline to design a matching algorithm: under such an algorithm, the empirical proportion of draws of the arm $a^*$ with smallest mean should converge to 1. The literature on regret minimization in bandit models (see [2] for a survey) provides candidate algorithms that have this type of behavior, and we propose to use the

Thompson Sampling (TS) algorithm [1, 22]. Given independent prior distribution on the mean of each arm, this Bayesian algorithm selects an arm at random according to its posterior probability of being optimal (in our case, the arm with smallest mean). Letting $\pi_a^t$ refer to the posterior distribution of $\mu_a$ after $t$ samples, this can be implemented as

$$\texttt{TS: Sample } \forall a \in \{1, \ldots, K\}, \theta_a(t) \sim \pi_a^{t-1}, \text{ then play } A_t = \arg\min_{a \in \{1, \ldots, K\}} \theta_a(t).$$

It follows from Theorem 12 in Appendix 5 that if Thompson Sampling is run without stopping, $N_{a^*}(t)/t$ converges almost surely to 1, for every $\boldsymbol{\mu}$. As TS is an anytime sampling strategy (i.e. that does not depend on $\delta$), Lemma 4 below permits to justify that on every instance of $\mathcal{H}_<$ with a unique optimal arm, under this algorithm $\tau_{\text{Box}} \simeq (1/d(\mu_1, \theta)) \ln(1/\delta)$. However, TS cannot be optimal for $\boldsymbol{\mu} \in \mathcal{H}_>$, as the empirical proportions of draws cannot converge to $\boldsymbol{w}^*(\boldsymbol{\mu}) \neq \mathbf{1}_{a^*}$.

To summarize, we presented a simple stopping rule, $\tau_{\text{Box}}$, that can be asymptotically optimal for every $\boldsymbol{\mu} \in \mathcal{H}_<$ if it is used in combination with Thompson Sampling and for $\boldsymbol{\mu} \in \mathcal{H}_>$ if it is used in combination with LCB. But neither of these two sampling rules are good for the other type of instances, which is a big limitation for a practical use of either of these. In the next section, we propose a new Thompson Sampling like algorithm that ensures the right exploration under both $\mathcal{H}_<$ and $\mathcal{H}_>$. In Section 5, we further present an improved stopping rule that may stop significantly earlier than $\tau_{\text{Box}}$ on instances of $\mathcal{H}_<$, by aggregating samples from multiple arms that look small.

We now argue that ensuring the sampling proportions converge to $\boldsymbol{w}^*$ is sufficient for reaching the optimal sample complexity, at least in an asymptotic sense. The proof can be found in Appendix C.

**Lemma 4.** *Fix $\boldsymbol{\mu} \in \mathcal{H}$. Fix an anytime sampling strategy $(A_t)$ ensuring $\frac{N_t}{t} \to \boldsymbol{w}^*(\boldsymbol{\mu})$. Let $\tau_\delta$ be a stopping rule such that $\tau_\delta \leq \tau_\delta^{\text{Box}}$, for a Box stopping rule (2) whose threshold functions $C_\lessgtr$ satisfy the following: they are non-decreasing in $r$ and there exists a function $f$ such that,*

$$\forall r \geq r_0, \ C_\lessgtr(\delta, r) \leq f(\delta) + \ln r, \quad \text{where} \quad f(\delta) = \ln(1/\delta) + o(\ln(1/\delta)).$$

*Then $\limsup_{\delta \to 0} \frac{\tau_\delta}{\ln \frac{1}{\delta}} \leq T^*(\boldsymbol{\mu})$ almost surely.*

## 4 Murphy Sampling

In this section we denote by $\Pi_n = \mathbb{P}(\cdot | \mathcal{F}_n)$ the posterior distribution of the mean parameters after $n$ rounds. We introduce a new (randomised) sampling rule called *Murphy Sampling* after Murphy's Law, as it performs some conditionning to the "worst event" ($\boldsymbol{\mu} \in \mathcal{H}_<$):

$$\texttt{MS: Sample } \boldsymbol{\theta}_t \sim \Pi_{t-1}(\cdot | \mathcal{H}_<), \text{ then play } A_t = a^*(\boldsymbol{\theta}_t). \tag{4}$$

As we will argue below, the subtle difference of sampling from $\Pi_{n-1}(\cdot | \mathcal{H}_<)$ instead of $\Pi_{n-1}$ (regular Thompson Sampling) ensures the required split personality behavior (see Lemma 1). Note that MS always conditions on $\mathcal{H}_<$ (and never on $\mathcal{H}_>$) regardless of the position of $\boldsymbol{\mu}$ w.r.t. $\gamma$. This is different from the symmetric Top Two Thompson Sampling [29], which essentially conditions on $a^*(\boldsymbol{\theta}) \neq a^*(\boldsymbol{\mu})$ a fixed fraction $1 - \beta$ of the time, where $\beta$ is a parameter that needs to be tuned with knowledge of $\boldsymbol{\mu}$. MS on the other hand needs no parameters.

Also note that MS is an anytime sampling algorithm, being independent of the confidence level $1 - \delta$. The confidence will manifest only in the stopping rule.

MS is technically an instance of Thompson Sampling with a joint prior $\Pi$ supported only on $\mathcal{H}_<$. This viewpoint is conceptually funky, as we will apply MS identically to $\mathcal{H}_<$ *and* $\mathcal{H}_>$. To implement MS, we use that independent conjugate per-arm priors induce likewise posteriors, admitting efficient (unconditioned) posterior sampling. Rejection sampling then achieves the required conditioning. Its computational cost is limited: the acceptance probability cannot be much smaller than the risk $\delta$ provided to the algorithm. Indeed, the fact that the stopping rule (see Section 5) has not yet fired, combined with the posterior concentration (Proposition 6) and the convergence of the sampling efforts to track the sampling proportions (Theorem 5) reveals that the MS rejection sampling step accepts with probability at least of order $\delta/(\ln t)^3$. So for reasonable values of $\delta$, this can be small and require a few thousands of draws (not a big deal for today's computers), but it cannot be *prohibitively* small.

The rest of this section is dedicated to the analysis of MS. First, we argue that the MS sampling proportions converge to the oracle weights of Lemma 1.

**Assumption** For purpose of analysis, we need to assume that the parameter space $\Theta \ni \boldsymbol{\mu}$ (or the support of the prior) is the interior of a bounded subset of $\mathbb{R}^K$. This ensures that $\sup_{\mu,\theta\in\Theta} d(\mu,\theta) < \infty$ and $\sup_{\mu,\theta\in\Theta} \|\mu-\theta\| < \infty$. This assumption is common [16, Section 7.1], [29, Assumption 1]. We also assume that the prior $\Pi$ has a density $\pi$ with bounded ratio $\sup_{\mu,\theta\in\Theta} \frac{\pi(\theta)}{\pi(\mu)} < \infty$.

**Theorem 5.** *Under the above assumption, MS ensures $\frac{N_t}{t} \to \boldsymbol{w}^*(\boldsymbol{\mu})$ a.s. for any $\boldsymbol{\mu} \in \mathcal{H}$.*

We give a sketch of the proof below, the detailed argument can be found in Appendix D, Theorems 12 and 13. Given the convergence of the weights, the asymptotic optimality in terms of sample complexity follows by Lemma 4, if MS is used with an appropriate stopping rule (Box (2) or the improved Aggregate stopping rule discussed in Section 5).

**Proof Sketch** First, consider $\boldsymbol{\mu} \in \mathcal{H}_<$. In this case the conditioning in MS is asymptotically immaterial as $\Pi_n(\mathcal{H}_<) \to 1$, and the algorithm behaves like regular Thompson Sampling. As Thompson sampling has sublinear pseudo-regret [1], we must have $\mathbb{E}[N_1(t)]/t \to 1$. The crux of the proof in the appendix is to show the convergence occurs almost surely.

Next, consider $\boldsymbol{\mu} \in \mathcal{H}_>$. Following [29], we denote the sampling probabilities in round $n$ by $\psi_a(n) = \Pi_{n-1}\left(a = \arg\min_j \theta_j | \mathcal{H}_<\right)$, and abbreviate $\Psi_a(n) = \sum_{t=1}^n \psi_a(t)$ and $\bar\psi_a(n) = \Psi_a(n)/n$. The main intuition is provided by

**Proposition 6** ([29, Proposition 4]). *For any open subset $\tilde\Theta \subseteq \Theta$, the posterior concentrates at rate $\Pi_n(\tilde\Theta) \doteq \exp\left(-n\min_{\boldsymbol{\lambda}\in\tilde\Theta} \sum_a \bar\psi_a(n) d(\mu_a, \lambda_a)\right)$ a.s. where $a_n \doteq b_n$ means $\frac{1}{n}\ln\frac{a_n}{b_n} \to 0$.*

Let us use this to analyze $\psi_a(n)$. As we are on $\mathcal{H}_>$, the posterior $\Pi_n(\mathcal{H}_<) \to 0$ vanishes. Moreover, $\Pi_n\left(a = \arg\min_j \theta_j, \mathcal{H}_<\right) \sim \Pi_n(\theta_a < \gamma)$ as the probability that multiple arms fall below $\gamma$ is negligible. Hence

$$\psi_a(n+1) \;\sim\; \frac{\Pi_n(\mu_a < \gamma)}{\sum_j \Pi_n(\mu_j < \gamma)} \;\doteq\; \frac{\exp\left(-n\bar\psi_a(n) d(\mu_a, \gamma)\right)}{\sum_j \exp\left(-n\bar\psi_j(n) d(\mu_j, \gamma)\right)}.$$

To get a good sense for what this means, let's analyse the version with equality. Using that $w_a^* d(\mu_a, \gamma)$ is constant (Lemma 1), we see

$$\psi_a(n+1) \;\leq\; e^{-n(\bar\psi_a(n) - w_a^*) d(\mu_a, \gamma)}.$$

Now this means that whenever $\bar\psi_a(n) \geq w_a^* + \epsilon$, we find that $\psi_a(n+1) \leq e^{-n\epsilon d_a} \approx 0$ is exponentially small, and hence $\bar\psi_a(n+1) \approx \frac{n}{n+1}\bar\psi_a(n)$ decays hyperbolically (i.e. without lower bound). Hence $\limsup_{n\to\infty} \bar\psi_a(n) \leq w_a^* + \epsilon$. As this holds for all arms $a$ and $\epsilon > 0$, we must have $\lim_n \psi_a(n) = w_a^*$.

## 5 Improved Stopping Rule and Confidence Intervals

Theorem 7 below provides a new self-normalized deviation inequality that given a *subset* of arms controls uniformly over time how the *aggregated mean* of the samples obtained from those arms can deviate from the smallest (resp. largest) mean in the subset. More formally for $\mathcal{S} \subseteq [K]$, we introduce

$$N_{\mathcal{S}}(t) \;=\; \sum_{a\in\mathcal{S}} N_a(t) \qquad \text{and} \qquad \hat\mu_{\mathcal{S}}(t) \;=\; \frac{\sum_{a\in\mathcal{S}} N_a(t)\hat\mu_a(t)}{N_{\mathcal{S}}(t)}$$

and recall $d^+(u,v) = d(u,v)\mathbb{1}_{(u\leq v)}$ and $d^-(u,v) = d(u,v)\mathbb{1}_{(u\geq v)}$. We prove the following for one-parameter exponential families.

**Theorem 7.** *Let $T : \mathbb{R}^+ \to \mathbb{R}^+$ be the function defined by*

$$T(x) \;=\; 2h^{-1}\left(1 + \frac{h^{-1}(1+x) + \ln\zeta(2)}{2}\right) \tag{5}$$

*where $h(u) = u - \ln(u)$ for $u \geq 1$ and $\zeta(s) = \sum_{n=1}^\infty n^{-s}$. For every subset $\mathcal{S}$ of arms and $x \geq 0.04$,*

$$\mathbb{P}\left(\exists t \in \mathbb{N} : N_{\mathcal{S}}(t)d^+\left(\hat\mu_{\mathcal{S}}(t), \min_{a\in\mathcal{S}}\mu_a\right) \geq 3\ln(1 + \ln(N_{\mathcal{S}}(t))) + T(x)\right) \;\leq\; e^{-x}, \tag{6}$$

$$\mathbb{P}\left(\exists t \in \mathbb{N} : N_{\mathcal{S}}(t)d^-\left(\hat\mu_{\mathcal{S}}(t), \max_{a\in\mathcal{S}}\mu_a\right) \geq 3\ln(1 + \ln(N_{\mathcal{S}}(t))) + T(x)\right) \;\leq\; e^{-x}. \tag{7}$$

The proof of this theorem can be found in Section F and is sketched below. It generalizes in several directions the type of results obtained by [18, 23] for Gaussian distributions and $|\mathcal{S}| = 1$. Going beyond subsets of size 1 will be crucial here to obtain better confidence intervals on minimums, or stop earlier in tests. Note that the threshold function $T$ introduced in (5) does not depend on the cardinality of the subset $\mathcal{S}$ to which the deviation inequality is applied. Tight upper bounds on $T$ can be given using Lemma 21 in Appendix F.3, which support the approximation $T(x) \simeq x + 3\ln(x)$.

## 5.1 An Improved Stopping Rule

Fix a subset prior $\pi : \wp(\{1, \ldots, K\}) \to \mathbb{R}^+$ such that $\sum_{\mathcal{S} \subseteq \{1, \ldots, K\}} \pi(\mathcal{S}) = 1$ and let $T$ be the threshold function defined in Theorem 7. We define the stopping rule $\tau^\pi := \tau_> \wedge \tau_<^\pi$, where

$$
\begin{aligned}
\tau_> &= \inf\left\{t \in \mathbb{N}^* : \forall a \in \{1, \ldots, K\} N_a(t)d^-(\hat{\mu}_a(t), \gamma) \geq 3\ln(1 + \ln(N_a(t))) + T(\ln(1/\delta))\right\}, \\
\tau_<^\pi &= \inf\left\{t \in \mathbb{N}^* : \exists \mathcal{S} : N_{\mathcal{S}}(t)d^+(\hat{\mu}_{\mathcal{S}}(t), \gamma) \geq 3\ln(1 + \ln(N_{\mathcal{S}}(t))) + T(\ln(1/(\delta\pi(\mathcal{S}))))\right\}.
\end{aligned}
$$

The associated recommendation rule selects $\mathcal{H}_>$ if $\tau^\pi = \tau_>$ and $\mathcal{H}_<$ if $\tau^\pi = \tau_<^\pi$. For the practical computation of $\tau_<^\pi$, the search over subsets can be reduced to nested subsets including arms sorted by increasing empirical mean and smaller than $\gamma$.

**Lemma 8.** *Any algorithm using the stopping rule $\tau^\pi$ and selecting $\hat{m} => $ iff $\tau^\pi = \tau_>$, is $\delta$-correct.*

From Lemma 8, proved in Appendix G, the prior $\pi$ doesn't impact the correctness of the algorithm. However it may impact its sample complexity significantly. First it can be observed that picking $\pi$ that is uniform over subset of size 1, i.e. $\pi(\mathcal{S}) = K^{-1}\mathbb{1}(|\mathcal{S}| = 1)$, one obtain a $\delta$-correct $\tau_{\text{Box}}$ stopping rule with thresholds functions satisfying the assumptions of Lemma 4. However, in practice (especially more moderate $\delta$), it may be more interesting to include in the support of $\pi$ subsets of larger sizes, for which $N_{\mathcal{S}}(t)d^+(\hat{\mu}_{\mathcal{S}}(t), \gamma)$ may be larger. We advocate the use of $\pi(\mathcal{S}) = K^{-1}\binom{K}{|\mathcal{S}|}^{-1}$, that puts the same weight on the set of subsets of each possible size.

**Links with Generalized Likelihood Ratio Tests (GLRT).** Assume we want to test $\mathcal{H}_0$ against $\mathcal{H}_1$ for composite hypotheses. A GLRT test based on $t$ observations whose distribution depends on some parameter $x$ rejects $\mathcal{H}_0$ if the test statistic $\max_{x \in \mathcal{H}_1} \ell(X_1, \ldots, X_t; x)/\max_{x \in \mathcal{H}_0 \cup \mathcal{H}_1} \ell(X_1, \ldots, X_t; x)$ has large values (where $\ell(\cdot; x)$ denotes the likelihood of the observations under the model parameterized by $x$). In our testing problem, the GLRT statistic for rejecting $\mathcal{H}_<$ is $\min_a N_a(t)d^-(\hat{\mu}_a(t), \gamma)$ hence $\tau_>$ is very close to a sequential GLRT test. However, the GLRT statistic for rejecting $\mathcal{H}_>$ is $\sum_{a=1}^K N_a(t)d^+(\hat{\mu}_a(t), \gamma)$, which is quite different from the stopping statistic used by $\tau_<^\pi$. Rather than *aggregating samples* from arms, the GLRT statistic is *summing evidence* for exceeding the threshold. Using similar martingale techniques as for proving Theorem 7, one can show that replacing $\tau_<^\pi$ by

$$
\tau_<^{\text{GLRT}} = \inf\left\{t \in \mathbb{N}^* : \sum_{a:\hat{\mu}_a(t) \leq \gamma} [N_a(t)d^+(\hat{\mu}_a(t), \gamma) - 3\ln(1 + \ln(N_a(t)))]^+ \geq KT\left(\frac{\ln(1/\delta)}{K}\right)\right\}
$$

also yields a $\delta$-correct algorithm (see [21])[1]. At first sight, $\tau_<^\pi$ and $\tau_<^{\text{GLRT}}$ are hard to compare: the stopping statistic used by the latter can be larger than that used by the former, but it is compared to a smaller threshold. In Section 6 we will provide empirical evidence in favor of aggregating samples.

## 5.2 A Confidence Intervals Interpretation

Inequality (6) (and a union bound over subsets) also permits building a tight upper confidence bound on the minimum $\mu^*$. Indeed, defining

$$
\text{U}_{\min}^\pi(t) := \max\left\{q : \max_{\mathcal{S} \subseteq \{1, \ldots, K\}} [N_{\mathcal{S}}(t)d^+(\hat{\mu}_{\mathcal{S}}(t), q) - 3\ln(1 + \ln(1 + N_{\mathcal{S}}(t)))] \leq T\left(\ln\frac{1}{\delta\pi(\mathcal{S})}\right)\right\},
$$

it is easy to show that $\mathbb{P}(\forall t \in \mathbb{N}, \mu^* \leq \text{U}_{\min}^\pi(t)) \geq 1 - \delta$. For general choices of $\pi$, this upper confidence bound may be much smaller than the naive bound $\min_a \text{U}_a(t)$ which corresponds to choosing $\pi$ uniform over subset of size 1. We provide an illustration supporting this claim in Figure 2

below. The two type of upper confidence bounds (Aggregate corresponding to $\pi(\mathcal{S}) = K^{-1}\binom{K}{|\mathcal{S}|}^{-1}$ and Box corresponding to $\pi(\mathcal{S}) = K^{-1}\mathbb{1}_{(|\mathcal{S}|=1)}$) are compared under uniform sampling in a Bernoulli bandit model that has $k$ arms with mean $0.1$ plus 4 arms with means $[0.2\ 0.3\ 0.4\ 0.5]$. The larger the number of arms close to minimum (here equal to it) is, the more UCB Aggregate beats UCB Box. Observe that using inequality (7) in Theorem 7 similarly allows to derive tighter lower confidence bounds on the maximum of several means.

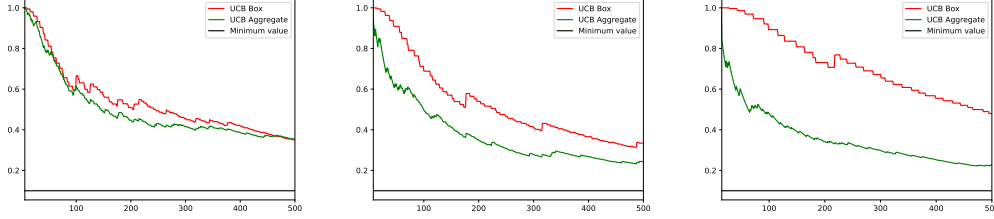

Figure 2: Illustration of the Box versus Aggregate Upper Confidence Bounds as a function of time on Bernoulli instance for $k = 1$ (left), $k = 3$ (middle) and $k = 10$ (right) minimal arms.

### 5.3 Sketch of the Proof of Theorem 7

Fix $\eta \in [0, 1 + e[$. Introducing $X_\eta(t) = [N_\mathcal{S}(t)d^+\left(\hat{\mu}_\mathcal{S}(t), \min_{a\in\mathcal{S}}\mu_a\right) - 2(1+\eta)\ln\left(1 + \ln N_\mathcal{S}(t)\right)]$, the cornerstone of the proof (Lemma 17) consists in proving that for all $\lambda \in [0, (1+\eta)^{-1}[$, there exists a martingale $M_t^\lambda$ that "almost" upper bounds $e^{\lambda X_\eta(t)}$: there exists a function $g_\eta$ such that

$$\mathbb{E}[M_0^\lambda] = 1 \quad \text{and} \quad \forall t \in \mathbb{N}^*, M_t^\lambda \geq e^{\lambda X_\eta(t) - g_\eta(\lambda)}. \tag{8}$$

From there, the proof easily follows from a combination of Chernoff method and Doob inequality:

$$\mathbb{P}\left(\exists t \in \mathbb{N}^* : X_\eta(t) > u\right) \quad \leq \quad \mathbb{P}\left(\exists t \in \mathbb{N}^* : M_t^\lambda > e^{\lambda u - g_\eta(\lambda)}\right) \leq \exp\left(-[\lambda u - g_\eta(\lambda)]\right).$$

Inequality (6) is then obtained by optimizing over $\lambda$, carefully picking $\eta$ and inverting the bound.

The interesting part of the proof is to actually build a martingale satisfying (8). First, using the so-called method of mixtures [6] and some specific fact about exponential families already exploited by [3], we can prove that there exists a martingale $\tilde{W}_t^x$ such that for some function $f$ (see Equation (14))

$$\{X_\eta(t) - f(\eta) \geq x\} \subseteq \left\{\tilde{W}_t^x \geq e^{\frac{x}{1+\eta}}\right\}.$$

From there it follows that, for every $\lambda$ and $z > 1$, $\left\{e^{\lambda(X_\eta(t) - f(\eta))} \geq z\right\} \subseteq \left\{e^{-\frac{\ln(z)}{\lambda(1+\eta)}}\tilde{W}_t^{\frac{1}{\lambda}\ln(z)} \geq 1\right\}$ and the trick is to introduce another mixture martingale,

$$\overline{M}_t^\lambda = 1 + \int_1^\infty e^{-\frac{\ln(z)}{\lambda(1+\eta)}}\tilde{W}_t^{\frac{1}{\lambda}\ln(z)}dz,$$

that is proved to satisfy $\overline{M}_t^\lambda \geq e^{\lambda[X_\eta(t) - f(\eta)]}$. We let $M_t^\lambda = \overline{M}_t^\lambda / \mathbb{E}[\overline{M}_t^\lambda]$.

## 6 Experiments

We discuss the results of numerical experiments performed on Gaussian bandits with variance 1, using the threshold $\gamma = 0$. Thompson and Murphy sampling are run using a flat (improper) prior on $\mathbb{R}$, which leads to a conjugate Gaussian posterior. The experiments demonstrate the flexibility of our MS sampling rule, which attains optimal performance on instances from both $\mathcal{H}_<$ and $\mathcal{H}_>$. Moreover, they show the advantage of using a stopping rule aggregating samples from subsets of arms when $\mu \in \mathcal{H}_<$. This aggregating stopping rule, that we refer to as $\tau^{\text{Agg}}$ is an instance of the $\tau^\pi$ stopping rule presented in Section 5 for $\pi(\mathcal{S}) = K^{-1}\binom{K}{|\mathcal{S}|}^{-1}$. We investigate the combined use of three sampling rules, MS, LCB and Thompson Sampling with three stopping rules, $\tau^{\text{Agg}}$, $\tau^{\text{Box}}$ and $\tau^{\text{GLRT}}$.

We first study an instance $\boldsymbol{\mu} \in \mathcal{H}_<$ with $K = 10$ arms that are linearly spaced between $-1$ and $1$. We run the different algorithms (excluding the TS sampling rule, that essentially coincides with MS on $\mathcal{H}_<$) for different values of $\delta$ and report the estimated sample complexity in Figure 3 (left). For each sampling rule, it appears that $\mathbb{E}[\tau^{\mathrm{Agg}}] \leq \mathbb{E}[\tau^{\mathrm{Box}}] \leq \mathbb{E}[\tau^{\mathrm{GLRT}}]$. Moreover, for each stopping rule MS is outperforming LCB, with a sample complexity of order $T^*(\boldsymbol{\mu}) \ln(1/\delta) + C$. Then we study an instance $\boldsymbol{\mu} \in \mathcal{H}_>$ with $K = 5$ arms that are linearly spaced between $0.5$ and $1$, with $\tau^{\mathrm{Agg}}$ as the sampling rule (which matters little as the algorithm mostly stops because of $\tau_>$ on $\mathcal{H}_>$). Results are reported in Figure 3 (right), in which we see that MS is performing very similarly to LCB (that is also proved optimal on $\mathcal{H}_>$), while vanilla TS fails dramatically. On those experiments, the empirical error was always zero, which shows that our theoretical thresholds are still quite conservative. More experimental results can be found in Appendix A: an illustration of the convergence properties of the MS sampling rule as well as a larger-scale comparison of stopping rules under $\mathcal{H}_<$.

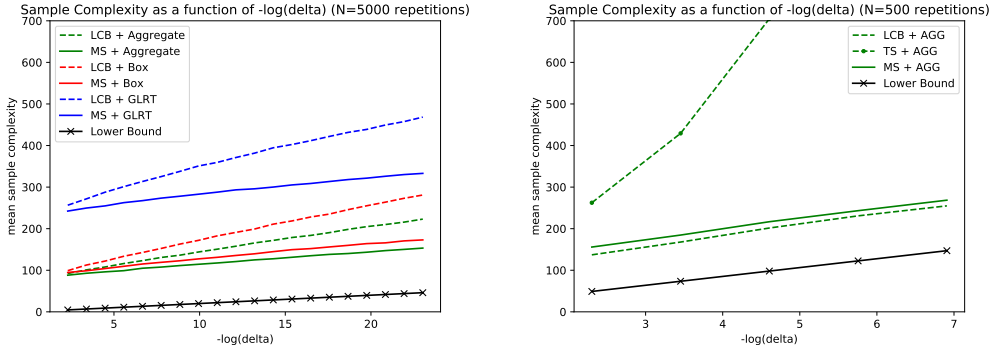

Figure 3: $\mathbb{E}[\tau_\delta]$ as a function of $\ln(1/\delta)$ for several algorithms on an instance $\mu \in \mathcal{H}_<$ (left) and $\mu \in \mathcal{H}_>$ (right), estimated using $N = 5000$ (resp. 500) repetitions.

## 7 Discussion

We propose new sampling and stopping rules for sequentially testing the minimum of means. As our guiding principle, we first prove sample complexity lower bounds, characterized the emerging oracle sample allocation $\boldsymbol{w}^*$, and develop the Murphy Sampling strategy to match it asymptotically. We observe in the experiments that the asymptotic regime does not necessarily kick in at moderate confidence $\delta$ (Figure 4, left) and that there is an important lower-order term to the practical sample complexity (Figure 3). It is an intriguing open problem of theoretical and practical importance to characterize and match optimal behavior at moderate confidence. We make first contributions in both directions: we prove tighter sample complexity lower bounds for symmetric algorithms (Proposition 2, Theorem 3) and we design aggregating confidence intervals which are tighter in practice (Figure 2).

The importance of this perspective arises, as highlighted in the introduction, from the *hierarchical* application of maxima/minima in learning applications. A better understanding of the moderate confidence regime for learning minima will very likely translate into new insights and methods for learning about hierarchical structures, where the benefits accumulate with depth.

## Footnotes

[1]In fact, we can slightly sharpen the bound by observing that we are controlling the deviation of a single composite arm, allowing us to replace (5) by $T(x) = 2h^{-1}\left(1 + \frac{x + \ln \zeta(2)}{2}\right)$, see [21, Appendix A.1]

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
