[Supplementary Material · full.pdf]

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

[2] As it is written this inequality is not actually correct: in the definition of the confidence interval for arm $a$, $\ln(1/\delta)$ should be replaced by $\ln(1/\delta) + c \ln \ln(1/\delta) + d \ln(1 + \ln(N_a(t)))$ for some constants $c$ and $d$ (see the discussion in Section 3.2). However, the reasoning that we present with the stylized confidence intervals can be adapted to handle those correct threshold functions, at the price of extra technicalities (e.g., Lemma 22 in [23]).

[3]In [21, Theorem 16] we omit this inequality and sharpen the threshold to $T(x) = 2\tilde{h}\left(\frac{h^{-1}(x)+\ln\zeta(2)}{2}\right)$ for $x \ge 3.81$, where $\tilde{h}(x) = h^{-1}(x)e^{1/h^{-1}(x)} \le h^{-1}(1+x)$.

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

# A  Additional Experimental Results

We first report in Figure 4 further results regarding the convergence of the sampling proportions $N_a(\tau)/\tau$ under the two instances of $\mathcal{H}_<$ and $\mathcal{H}_>$ described in Section 6, for the smallest value of $\delta$ used in each experiment and under the stopping rule $\tau^{\text{Agg}}$. Under $\mathcal{H}_<$ we see that MS has indeed spent a larger fraction of the time on the optimal arm, even if it does not yet reach the fraction 1 prescribed by the lower bound. One can also note that the empirical proportions of draws of the arms under LCB are very close to the sub-optimal weights obtained in Proposition 16 in Appendix E, which are added to the plot. Under $\mathcal{H}_>$, we see that the empirical fractions of draws of both MS and LCB converge to $\boldsymbol{w}^*(\boldsymbol{\mu})$ whereas the TS sampling rule departs significantly from those optimal weights, by drawing mostly arm 1.

Then we go deeper into investigating the impact of the proposed sampling rule under instances of $\mathcal{H}_<$. Indeed, we expect that grouping samples from several arms will help stop earlier as the number of arms under the threshold $\gamma$ increases, which we illustrate with the following experiment. Consider $K = 100$ Gaussian arms with variance 1 and $\gamma = 0$. For several values of $k \in \{1, \ldots, K\}$, we consider an instance in which there are $k$ arms with mean $-1$ and $K - k$ arms with mean 0. Note that all those instances have the same (asymptotic) theoretical sample complexity, which is $T^*(\boldsymbol{\mu}) \ln(1/\delta)$, but in a regime with "large" $\delta$ (here we take $\delta = 0.1$), we expect this aggregating of samples to reduce significantly the sample complexity especially when there are a lot of arms below $\gamma$.

Figure 5 (left) reports the sample complexity of the Agg, Box and GLRT stopping rule, each used in combination with either the LCB or the MS sampling rule, for different values of $k$. On can note first that for a given stopping rule, MS is always outperforming LCB. Then, $\tau^{\text{Agg}}$ outperforms $\tau^{\text{Box}}$ for all the values of $k$, as well as $\tau^{\text{GLRT}}$ for values that are smaller than 70. GLRT is thus a better candidate only when the number of arms below the threshold is very large. This may be explained by the support plot displayed in Figure 5 (right): for each value of $k$, we report the number of arms in the subset $\mathcal{S}$ that was used for stopping, that is which satisfies $N_\mathcal{S}(\tau)d^+(\hat{\mu}_\mathcal{S}(\tau), \gamma) \geq \ln(1 + \ln(N_\mathcal{S}(\tau))) + \ln(1/(\delta\pi(\mathcal{S})))$ in the case of Box and Agg. For the GLRT, the support is the number of arms for which $\hat{\mu}_a(\tau) \leq \gamma$, whose evidence for being below the threshold is included in the definition of the GLRT statistic. The support plot highlights that GLRT may sum evidence from more arms than the number of arms whose samples are aggregated by Agg, and in a regime in which the thresholds to which the two stopping statistics are compared are similar, this may favor GLRT. In Figure 6, we report similar experiments in instances in which $K = 100$ and for each $k$ there are $k$ linearly spaced arms below the threshold and $K - k$ arms with mean 0. In that case, even for large values of $k$, GLRT does not outperform the Aggregating stopping rule, which successfully combines samples from several arms below the threshold with different means.

# B  Proofs for the Sample Complexity Lower Bounds

We first need the following Lemma, that tells us that a $\delta$-correct strategy stops with probability at most $2\delta$ if all arms have mean exactly $\gamma$.

**Lemma 9.** *Let* $\boldsymbol{\gamma} = (\gamma, \ldots, \gamma)$. *For any $\delta$-correct test,* $\mathbb{P}_{\boldsymbol{\gamma}}[\tau < \infty] \leq 2\delta$.

*Proof.* Let $m > 0$, $\epsilon > 0$, $\boldsymbol{\mu} = (\gamma + \epsilon, \ldots, \gamma + \epsilon)$ and $\boldsymbol{\mu}' = (\gamma - \epsilon, \ldots, \gamma - \epsilon)$. Then the informational inequality of [23, Lemma 1] applied to the event $\{\tau \leq m, \hat{m} = >\}$, followed by $\text{kl}(p, q) \geq 2(p - q)^2$, implies that

$$
\begin{aligned}
md(\gamma + \epsilon, \gamma - \epsilon) &\geq \text{kl}\big(\mathbb{P}_{\boldsymbol{\mu}}(\tau \leq m, \hat{m} = >), \mathbb{P}_{\boldsymbol{\mu}'}(\tau \leq m, \hat{m} = >)\big) \\
&\geq 2\big(\mathbb{P}_{\boldsymbol{\mu}}(\tau \leq m, \hat{m} = >) - \mathbb{P}_{\boldsymbol{\mu}'}(\tau \leq m, \hat{m} = >)\big)^2 \\
&= 2\big(\mathbb{P}_{\boldsymbol{\mu}}(\tau \leq m) - \mathbb{P}_{\boldsymbol{\mu}}(\tau \leq m, \hat{m} = <) - \mathbb{P}_{\boldsymbol{\mu}'}(\tau \leq m, \hat{m} = >)\big)^2 \\
&\geq 2\big(\mathbb{P}_{\boldsymbol{\mu}}(\tau \leq m) - 2\delta\big)_+^2
\end{aligned}
$$

and thus

$$
\mathbb{P}_{\boldsymbol{\mu}}(\tau \leq m) \leq 2\delta + \sqrt{\frac{md(\gamma + \epsilon, \gamma - \epsilon)}{2}} \,.
$$

Figure 4: Empirical proportions of samples versus $w^*(\mu)$ for one instance in $\mathcal{H}_<$ (left) and one instance in $\mathcal{H}_>$ (right), in the same experimental setup as that of Figure 3.

Figure 5: Sample complexity (left) and support when stopping (right) for different algorithms as a function of the number $k$ of arms below the threshold $\gamma = 0$ on an instance for which $\mu_a \in \{-1, 0\}$.

Figure 6: Sample complexity (left) and support when stopping (right) for different algorithms as a function of the number $k$ of arms below the threshold $\gamma = 0$ on an instance for which the $k$ arms below $\gamma$ are linearly spaced between -1 and 0.

Letting $\epsilon$ go to 0, one gets $\mathbb{P}_{\gamma}(\tau \le m) \le 2\delta$ and thus

$$\mathbb{P}_{\gamma}(\tau < \infty) = \mathbb{P}_{\gamma}\left(\bigcup_{m>0}(\tau < m)\right) = \lim_{m\to\infty}\mathbb{P}_{\gamma}(\tau < m) \le 2\delta. \qquad \qquad \square$$

## B.1 Proof of Lemma 1

If $\mu^* < \gamma$, then we find

$$\frac{1}{T^*(\boldsymbol{\mu})} = \max_{\boldsymbol{w}\in\Delta}\sum_{a:\mu_a<\gamma} w_a d(\mu_a,\gamma) = \max_{a:\mu_a<\gamma} d(\mu_a,\gamma) = d(\mu^*,\gamma) \qquad \text{where} \qquad w_a^* = \mathbf{1}_{a=a^*}.$$

On the other hand, if $\mu^* > \gamma$, we find

$$\frac{1}{T^*(\boldsymbol{\mu})} = \max_{\boldsymbol{w}\in\Delta}\min_a w_a\, d(\mu_a,\gamma) = \frac{1}{\sum_a \frac{1}{d(\mu_a,\gamma)}} \qquad \text{where} \qquad w_a^* = \frac{\frac{1}{d(\mu_a,\gamma)}}{\sum_j \frac{1}{d(\mu_j,\gamma)}}.$$

## B.2 Proof of Proposition 2

Let $\boldsymbol{\gamma} = (\gamma,\dots,\gamma)$, and let $m > 0$. Fix $a \in \{1,\dots,K\}$. By Lemma 9,
$$\mathbb{E}_{\gamma}[N_a(\tau \wedge m)] \ge \mathbb{E}_{\gamma}[N_a(m)] - m\,\mathbb{P}_{\gamma}(\tau < m) \ge \mathbb{E}_{\gamma}[N_a(m)] - 2\delta\,m.$$
Then, by the informational lower bound (F-long) and by the generalized Pinsker inequality (Lemma 2 of [13]), one obtains

$$mk \ge \sum_{j=1}^{K}\mathbb{E}_{\gamma}[N_j(\tau \wedge m)]\,d(\mu_a,\gamma)$$

$$\ge \mathrm{kl}\left(\frac{\mathbb{E}_{\gamma}[N_a(\tau \wedge m)]}{m}, \frac{\mathbb{E}_{\boldsymbol{\mu}}[N_a(\tau \wedge m)]}{m}\right)$$

$$\ge \mathrm{kl}\left(\frac{1}{K} - 2\delta, \frac{\mathbb{E}_{\boldsymbol{\mu}}[N_a(\tau \wedge m)]}{m} \wedge \left(\frac{1}{K} - 2\delta\right)\right)$$

$$\ge \frac{K}{2}\left(\frac{1}{K} - 2\delta - \frac{\mathbb{E}_{\boldsymbol{\mu}}[N_a(\tau \wedge m)]}{m}\right)_+^2.$$

It follows that

$$\mathbb{E}_{\boldsymbol{\mu}}[N_a(\tau \wedge m)] \ge \frac{m}{K} - 2\delta m - m\sqrt{\frac{2mk}{K}},$$

and the result follows from the choice $m = 2K(1/K - 2\delta)^2/(9k)$.

## B.3 Proof of Theorem 3

By the informational lower bound (F-long) of [13],
$$\sum_a \mathbb{E}_{\boldsymbol{\mu}}[N_a(\tau)]\,d_+(\mu_a,\gamma) = \sum_{a:\mu_a<\gamma}\mathbb{E}_{\boldsymbol{\mu}}[N_a(\tau)]\,d(\mu_a,\gamma) \ge \mathrm{kl}(\delta, 1-\delta),$$
and by Proposition 2, for all $a \in \{1,\dots,K\}$,

$$\mathbb{E}_{\boldsymbol{\mu}}[N_a(\tau)] \ge n := \frac{2\left(1 - 2\delta K^3\right)}{27K^2 k}.$$

Hence,

$$\mathbb{E}_{\boldsymbol{\mu}}[\tau] = \sum_a \mathbb{E}_{\boldsymbol{\mu}}[N_a(\tau)] \ge \min\left\{\sum_{a=1}^{K} n_a \quad \text{such that} \sum_a n_a\,d_+(\mu_a,\gamma) \ge \mathrm{kl}(\delta,1-\delta) \text{ and } \forall a, n_a \ge n\right\}.$$

The solution of this minimization problem is: $n_a^* = n$ for all $a > 1$, and

$$n_1^* = \frac{\mathrm{kl}(\delta, 1-\delta) - n\sum_{a>1} d_+(\mu_a,\gamma)}{d(\mu_1,\gamma)}.$$

Thus,

$$\mathbb{E}_{\boldsymbol{\mu}}[\tau] \ge \sum_{a=1}^{K} n_a^* = \frac{\mathrm{kl}(\delta, 1-\delta)}{d(\mu_1,\gamma)} + n\sum_a\left(1 - \frac{d_+(\mu_a,\gamma)}{d(\mu_1,\gamma)}\right).$$

## C  Weight Convergence Implies Optimal Sample Complexity (Lemma 4)

Fix $\boldsymbol{\mu} \in \mathcal{H}_<$. Then there exists an event $\mathcal{E}$ such that $N_1(t)/t \to w_1^*(\boldsymbol{\mu})$ and $\hat{\mu}_1(t) \to \mu_1$. On this event $\mathcal{E}$, for all $\epsilon > 0$, there exists $t_0$ such that for $t \geq t_0$, $N_1(t)d(\hat{\mu}_1(t), \gamma) \geq (1 - \epsilon)td(\mu_1, \gamma)$. We use (2) to write

$$
\begin{aligned}
\tau_\delta &\leq & \tau_< &\leq \inf\{t \in \mathbb{N}^*, N_1(t)d^-(\hat{\mu}_1(t), \gamma) \geq C_<(\delta, N_1(t))\} \\
& & &\leq \inf\{t \geq t_0 : t(1 - \epsilon)d(\mu_1, \gamma) \geq C_<(\delta, t)\} \\
& & &\leq \inf\{t \geq t_0 : t(1 - \epsilon)d(\mu_1, \gamma) \geq f(\delta) + \ln(t)\}
\end{aligned}
$$

hence

$$
\tau_\delta \leq t_0 + \inf\left\{t \in \mathbb{N}^* : t \times [(1 - \epsilon)d(\mu_1, \gamma)] \geq \ln\left(\frac{t}{\delta}\right) + o(\ln(1/\delta))\right\}.
$$

Simple algebra (e.g. Lemma 22 in [23]) yields

$$
\tau_\delta \leq \frac{1}{(1 - \epsilon)d(\mu_1, \gamma)} \ln(1/\delta) + o(\ln(1/\delta))
$$

hence $\limsup_{\delta \to 0} \tau_\delta / \ln(1/\delta) \leq T^*(\boldsymbol{\mu})/(1 - \epsilon)$ for all $\epsilon$, thus $\limsup_{\delta \to 0} \tau_\delta / \ln(1/\delta) \leq T^*(\boldsymbol{\mu})$.

Fix $\boldsymbol{\mu} \in \mathcal{H}_>$. As each $a$ $w_a^*(\boldsymbol{\mu}) \neq 0$, all arms are drawn infinitely often, thus there exists an event $\mathcal{E}$ of probability 1 such that $N_a(t)/t \to w_a^*(\boldsymbol{\mu})$ and $\hat{\mu}_a(t) \to \mu_a$. On $\mathcal{E}$, for all $\epsilon > 0$, there exists $t_0$ such that for all $t \geq t_0$, $\forall a, N_a(t)d^-(\hat{\mu}_a(t), \gamma) \geq (1 - \epsilon)tw_a^*(\boldsymbol{\mu})d(\mu_a, \gamma)$. This time

$$
\begin{aligned}
\tau_\delta &\leq & \tau_> &= \inf\{t \in \mathbb{N}^* : \forall a, N_a(t)d^-(\hat{\mu}_a(t), \gamma) \geq C_>(\delta, N_a(t))\} \\
& & &\leq \inf\{t \in \mathbb{N}^* : \forall a, N_a(t)d^-(\hat{\mu}_a(t), \gamma) \geq C_>(\delta, t)\} \\
& & &\leq \inf\{t \geq t_0 : \forall a, (1 - \epsilon)tw_a^*(\boldsymbol{\mu})d(\mu_a, \gamma) \geq f(\delta) + \ln(t)\}
\end{aligned}
$$

and

$$
\tau_\delta \leq t_0 + \inf\left\{t \in \mathbb{N}^* : t \times \left[(1 - \epsilon)\min_a w_a^*(\boldsymbol{\mu})d(\mu_a, \gamma)\right] \geq \ln\left(\frac{t}{\delta}\right) + o(\ln(1/\delta))\right\}.
$$

Similarly one obtains

$$
\tau_\delta \leq \frac{T^*(\boldsymbol{\mu})}{(1 - \epsilon)} \ln(1/\delta) + o(\ln(1/\delta))
$$

and $\limsup_{\delta \to 0} \tau_\delta / \ln(1/\delta) \leq T^*(\boldsymbol{\mu})$.

## D  Analysis of Murphy Sampling (Proof of Theorem 5)

In this section we analyse the Murphy Sampling (4) sampling rule. Throughout we will make the assumption stated in Section 4.

Let $\Pi_n$ be the posterior on $\boldsymbol{\mu}$ after $n$ rounds. Let $\psi_a(t)$ denote the probability of sampling arm $a$ in round $t$, i.e.

$$
\psi_a(t) = \mathbb{P}(A_t = a | \mathcal{F}_{t-1}) = \Pi_{t-1}\left(a = \arg\min_j \mu_j \Big| \min_j \mu_j < \gamma\right).
$$

let $\Psi_a(n) = \sum_{t=1}^n \psi_a(t)$ and $\bar{\psi}_a(n) = \Psi_a(n)/n$. We will make use of the following result

**Proposition 10** ([29, Corollary 1]). *Let $\mathcal{S} \subseteq [K]$ be any subset of arms.*

$$
\sum_{a \in \mathcal{S}} \Psi_a(t) \to \infty \implies \lim_{t \to \infty} \frac{\sum_{a \in \mathcal{S}} N_a(t)}{\sum_{a \in \mathcal{S}} \Psi_a(t)} = 1 \qquad a.s.
$$

Our main assumption is the following (see e.g. [29, Proposition 3]). Let $\Theta_a \subseteq \mathbb{R}$ be an open set. Then

$$
\sup_t N_a(t) = \infty \implies \Pi_t(\theta_a \in \Theta_a) \to \mathbf{1}\{\mu_a \in \Theta_a\} \qquad a.s. \tag{9}
$$

$$
\sup_t N_a(t) < \infty \implies \inf_t \Pi_t(\theta_a \in \Theta_a) > 0 \qquad a.s. \tag{10}
$$

We first show that every arm is drawn infinitely often

**Proposition 11.** *Let $\boldsymbol{\mu} \in \mathcal{H}$ with $\mu^*$ not on the boundary of $\Theta_{a^*}$. Then the MS sampling rule ensures $N_a(t) \to \infty$ a.s. for all arm $a \in \{1, \dots, K\}$.*

*Proof.* By Proposition 10, it suffices to show $\Psi_a(t) \to \infty$. Toward contradiction assume that $\mathcal{A} \coloneqq \{a \mid \sup_t \Psi_a(t) < \infty\} \neq \varnothing$. Let $B = \{\theta \mid \theta < \mu^* - \epsilon\}$. Now for every arm $a \notin \mathcal{A}$, we have $\Pi_t(\theta_a \notin B) \to 1$ by (9). Let $C = \max_{a \in \mathcal{A}} \lim_t \Pi_t(\theta_a \in B)$. We have $C > 0$ by (10). But then

$$\sum_{a \in \mathcal{A}} \psi_a(t) \geq \Pi_t\left(\arg\min_a \theta_a \in \mathcal{A}, \min_a \theta_a < \gamma\right)$$

$$\geq \max_{a \in \mathcal{A}} \Pi_t(\theta_a \in B) \prod_{a \notin \mathcal{A}} \Pi_t(\theta_a \notin B) \to C > 0.$$

But this means that $\sum_{a \in \mathcal{A}} \Psi_a(t) \to \infty$, a contradiction. $\qquad\square$

The analysis now splits in 2 cases, depending on the location of $\min_a \mu_a$ w.r.t. $\gamma$. First we consider the case $\boldsymbol{\mu} \in \mathcal{H}_<$.

**Theorem 12.** *Consider $\boldsymbol{\mu} \in \mathcal{H}_<$ with minimal arms $\mathcal{A} = \{a \mid \mu_a = \mu_*\}$. Note that although Lemma 1 may not uniquely identify $\boldsymbol{w}^*(\boldsymbol{\mu})$, all candidate $\boldsymbol{w}^*(\boldsymbol{\mu})$ must satisfy $\sum_{a \in \mathcal{A}} w_a^*(\boldsymbol{\mu}) = 1$. The MS sampling rule ensures that the sampling frequencies converge to $\frac{\sum_{a \in \mathcal{A}} N_a(t)}{t} \to \sum_{a \in \mathcal{A}} w_a^*(\boldsymbol{\mu})$ a.s.*

*Proof.* Let $\zeta \in (\mu^*, \gamma \wedge \min_{a \notin \mathcal{A}} \mu_a)$. We have that

$$\sum_{a \in \mathcal{A}} \psi_a(t) \geq \Pi_t\left(\arg\min_j \mu_j \in \mathcal{A}, \min_j \mu_j \leq \gamma\right) \geq \max_{a \in \mathcal{A}} \underbrace{\Pi_t(\mu_a \leq \zeta)}_{\to 1 \text{ by } (9)} \prod_{a \notin \mathcal{A}} \underbrace{\Pi_t(\mu_a \geq \zeta)}_{\to 1 \text{ by } (9)} \to 1.$$

It follows that $\frac{\sum_{a \in \mathcal{A}} \Psi_a(t)}{t} \to 1$, and the result follows from Proposition 10. $\qquad\square$

Next we analyze the behavior of the MS sampling rule on $\boldsymbol{\mu} \in \mathcal{H}_>$. We follow the proof strategy of [29, Section G.1].

**Theorem 13.** *Let $\boldsymbol{\mu} \in \mathcal{H}_>$. Then $\frac{\boldsymbol{N}(t)}{t} \to \boldsymbol{w}^*(\boldsymbol{\mu})$ a.s.*

*Proof.* Let us abbreviate $\boldsymbol{w}^* \equiv \boldsymbol{w}^*(\boldsymbol{\mu})$. By Proposition 10, it suffices to show $\bar{\boldsymbol{\psi}}(t) \to \boldsymbol{w}^*$. We will show this by applying Proposition 14 below. First, recall from Lemma 1 that

$$T^*(\boldsymbol{\mu})^{-1} = \max_{\boldsymbol{w}} \min_{\boldsymbol{\lambda}:\min_a \lambda_a < \gamma} \sum_a w_a d(\mu_a, \lambda_a) = \max_{\boldsymbol{w}} \min_a w_a d(\mu_a, \gamma) = w_a^* d(\mu_a, \gamma) \quad \forall a. \quad (11)$$

Furthermore, by Proposition 6, for any $a \in [K]$

$$\Pi_n(\theta_a < \gamma) \doteq \exp\left(-n \min_{\boldsymbol{\lambda}:\lambda_a < \gamma} \sum_b \bar{\psi}_b(n) d(\mu_b, \lambda_b)\right) = \exp\left(-n\bar{\psi}_a(n) d(\mu_a, \gamma)\right).$$

In particular, there is a sequence $\epsilon_n$ decreasing to zero such that

$$\forall n: \quad \Pi_n(\theta_a < \gamma) \in \exp\left(-n\left(\bar{\psi}_a(n) d(\mu_a, \gamma) \pm \epsilon_n\right)\right).$$

To establish the precondition of Proposition 14 below, fix $a \in [K]$ and $c > 0$ and consider any round $n$ where $\bar{\psi}_a(n) \geq w_a^* + c$. Then

$$\psi_a(n) = \frac{\Pi_{n-1}\left(a = \arg\min_j \theta_j, \min_j \theta_j < \gamma\right)}{\Pi_{n-1}\left(\min_j \theta_j < \gamma\right)} \leq \frac{\Pi_{n-1}(\theta_a < \gamma)}{\max_a \Pi_{n-1}(\theta_a < \gamma)}$$

$$\leq \frac{e^{-n(\bar{\psi}_a(n) d(\mu_a, \gamma) - \epsilon_n)}}{\max_a e^{-n(\bar{\psi}_a(n) d(\mu_a, \gamma) + \epsilon_n)}} = e^{-n(\bar{\psi}_a(n) d(\mu_a, \gamma) - \min_a \bar{\psi}_a(n) d(\mu_a, \gamma) - 2\epsilon_n)}$$

By (11) $\min_a \bar{\psi}_a(n) d(\mu_a, \gamma) \leq \max_{\boldsymbol{w}} \min_a w_a d(\mu_a, \gamma) = w_a^* d(\mu_a, \gamma)$. Also $\bar{\psi}_a(n) \geq w_a^* + c$ so

$$\psi_a(n) \leq e^{-n\left((w_a^* + c) d(\mu_a, \gamma) - w_a^* d(\mu_a, \gamma) - 2\epsilon_n\right)} = e^{-n(cd(\mu_a, \gamma) - 2\epsilon_n)}.$$

Now as $\epsilon_n \to 0$, this establishes eventual exponential decay, hence ensuring that

$$\sum_n \psi_a(n) \mathbb{1}\left\{\bar{\psi}_a(n) \geq w_a^* + c\right\} < \infty$$

as required. The conclusion follows from Proposition 14.

$\qquad\square$

**Proposition 14** ([29, Simplified version of Lemma 11]). *Let $w^* \equiv w^*(\mu)$. Consider any sampling rule $(A_t)_t$. If for any arm $a \in [K]$ and all $c > 0$*

$$\sum_n \psi_a(n) \mathbb{1} \left\{ \bar{\psi}_a(n) \geq w_a^* + c \right\} < \infty$$

*then $\bar{\psi}(n) \to w^*$.*

# E  Analysis of LCB

The LCB algorithm (3) constructs confidence intervals $[\mathrm{L}_a(t), \mathrm{U}_a(t)]$. With the Box stopping rule (2) it stops and recommends $<$ when there exists $a$ such that $\mathrm{U}_a(t) < \gamma$. It stops and recommends $>$ when for all $a$, $\mathrm{L}_a(t) > \gamma$. When it has not stopped yet, it plays $A_{t+1} = \arg\min_a \mathrm{L}_a(t)$ the arm of smallest lower confidence bound.

In this section we show that LCB works fine on $\mathcal{H}_>$, but has the wrong behavior on $\mathcal{H}_<$. For simplicity we only consider the Gaussian case, in which confidence intervals have the stylized form

$$[\mathrm{L}_a(t), \mathrm{U}_a(t)] = \left[ \hat{\mu}_a(t) \mp \sqrt{\frac{2 \ln \frac{1}{\delta}}{N_a(t)}} \right].$$

Note that LCB is not anytime, as its sampling rule is also a function of the confidence level $1 - \delta$. We let $\tau_\delta$ denote the stopping rule associated to the algorithm that combines the LCB sampling rule with the Box stopping rule, both tuned for the confidence level $1 - \delta$.

Let $\mathcal{E}_\delta = \left\{ \forall t \forall a : |\mu_a - \hat{\mu}_a(t)| \leq \sqrt{\frac{2 \ln \frac{1}{\delta}}{N_a(t)}} \right\}$. By design of the confidence intervals $\mathbb{P}(\mathcal{E}_\delta^c) \leq \delta^2$. Moreover, on $\mathcal{E}_\delta$ the algorithm stops and outputs the correct recommendation.

We first show that LCB/Box is sample efficient on $\mathcal{H}_>$.

**Proposition 15.** *There is a function $\epsilon_\delta \to 0$ decreasing as $\delta \to 0$ such that for every $\mu \in \mathcal{H}_>$*

$$\lim_{\delta \to 0} \mathbb{P}_\mu \left( \frac{\tau_\delta}{\ln \frac{1}{\delta}} \leq (1 + \epsilon_\delta) T^*(\mu) \right) = 1.$$

*Proof.* Let $\kappa \in (0, 1)$. On the event $\mathcal{E}_{\delta^\kappa} \subseteq \mathcal{E}_\delta$ the algorithm (for confidence $\delta$) stops and outputs the correct recommendation, yielding

$$\forall a : \mathrm{L}_a(\tau) > \gamma.$$

Moreover, by the sampling rule we have $\forall a : \mathrm{L}_a(\tau) \approx \gamma$, and we will ignore the difference. We find

$$\gamma \approx \mathrm{L}_a(\tau) = \hat{\mu}_a(\tau) - \sqrt{\frac{2 \ln \frac{1}{\delta}}{N_a(\tau)}} \geq \mu_a - \sqrt{\kappa \frac{2 \ln \frac{1}{\delta}}{N_a(\tau)}} - \sqrt{\frac{2 \ln \frac{1}{\delta}}{N_a(\tau)}} = \mu_a - (1 + \sqrt{\kappa}) \sqrt{\frac{2 \ln \frac{1}{\delta}}{N_a(\tau)}}.$$

We conclude $N_a(\tau) \leq \frac{2(1+\sqrt{\kappa})^2 \ln \frac{1}{\delta}}{(\mu_a - \gamma)^2}$ and hence

$$\tau = \sum_a N_a(\tau) \leq (1 + \sqrt{\kappa})^2 \ln \frac{1}{\delta} \frac{2}{(\mu_a - \gamma)^2} = (1 + \sqrt{\kappa})^2 \ln \frac{1}{\delta} T^*(\mu).$$

The result follows by picking $\kappa = \frac{1}{\sqrt{-\ln \delta}}$, achieving $\kappa \to 0$ as $\delta \to 0$ yet $\mathbb{P}(\mathcal{E}_{\delta^\kappa}^c) \leq \delta^\kappa \to 0$. $\square$

Next we consider the behavior on $\mathcal{H}_<$. We characterize the inefficiency of LCB/Box on $\mathcal{H}_<$.

**Proposition 16.** *There is a function $\epsilon_\delta \to 0$ decreasing as $\delta \to 0$ such that for every bandit model $\mu \in \mathcal{H}_<$ with $\mu_1 < \gamma < \mu_2 \leq \ldots$ i.e. on which there is only a single arm below the threshold,*

$$\lim_{\delta \to 0} \mathbb{P}_\mu \left( \forall a \neq a^*(\mu) : \frac{N_a(\tau)}{\ln \frac{1}{\delta}} \geq (1 - \epsilon_\delta) \frac{2}{(\mu_a + \gamma - 2\mu_1)^2} \right) = 1.$$

*Proof.* Let $\kappa \in (0,1)$. We analyse the algorithm on the event $\mathcal{E}_{\delta^\kappa} \subseteq \mathcal{E}_\delta$, on which it stops and recommends the correct output. At that time $\tau$, we know

$$\mathrm{U}_1(\tau) \le \gamma \qquad \text{and also} \qquad \forall a : \mathrm{L}_1(\tau) \le \mathrm{L}_a(\tau).$$

Since we are on the event $\mathcal{E}_{\delta^\kappa}$, we know

$$\gamma \ge \mathrm{U}_1(\tau) = \hat{\mu}_1(\tau) + \sqrt{\frac{2\ln\frac{1}{\delta}}{N_1(\tau)}} \ge \mu_1 + \sqrt{\kappa\frac{2\ln\frac{1}{\delta}}{N_1(\tau)}} + \sqrt{\frac{2\ln\frac{1}{\delta}}{N_1(\tau)}} = \mu_1 + \left(1 + \sqrt{\kappa}\right)\sqrt{\frac{2\ln\frac{1}{\delta}}{N_1(\tau)}}.$$

On the other hand, we know for each other arm $a \ne 1$ that

$$\sqrt{\frac{2\ln\frac{1}{\delta}}{N_a(\tau)}} = \hat{\mu}_a(\tau) - \mathrm{L}_a(\tau) \le \hat{\mu}_a(\tau) - \mathrm{L}_1(\tau) \le \mu_a + \sqrt{\kappa\frac{2\ln\frac{1}{\delta}}{N_a(\tau)}} - \mathrm{L}_1(\tau).$$

Finally, since $\mathrm{L}_1(\tau) = \mathrm{U}_1(\tau) - 2\sqrt{\frac{2\ln\frac{1}{\delta}}{N_1(\tau)}}$ and $\mathrm{U}_1(\tau) \approx \gamma$ (we will ignore the difference), we find

$$\left(1 - \sqrt{\kappa}\right)\sqrt{\frac{2\ln\frac{1}{\delta}}{N_a(\tau)}} \le \mu_a - \gamma + 2\sqrt{\frac{2\ln\frac{1}{\delta}}{N_1(\tau)}} \le \mu_a - \gamma + 2\frac{\gamma - \mu_1}{1 + \sqrt{\kappa}}$$

All in all, this shows

$$N_a(\tau) \ge \frac{2(1 - \sqrt{\kappa})^2 \ln\frac{1}{\delta}}{\left(\mu_a - \gamma + 2\frac{\gamma - \mu_1}{1+\sqrt{\kappa}}\right)^2} \rightarrow \frac{2}{\left(\mu_a + \gamma - 2\mu_1\right)^2}.$$

The result follows by considering the sequence $\kappa$ exhibited in the proof of Proposition 15. $\qquad\square$

Now this demonstrates a problem, since Lemma 1 shows that optimal algorithms necessarily have $\frac{N_a(\tau)}{\ln\frac{1}{\delta}} \to 0$, but instead for LCB it tends to a specific positive constant. In other words, a non-vanishing hence significant portion of the samples are wasted "exploring" suboptimal arms.

## F  Proof of the Deviation Inequality (Theorem 7)

To ease the notation, we introduce $\mu_{\mathcal{S}}^{\min} = \min_{a\in\mathcal{S}} \mu_a$ and $\mu_{\mathcal{S}}^{\max} = \max_{a\in\mathcal{S}} \mu_a$. Fix $\eta > 0$ and $c > 0$ and define

$$\begin{aligned} X_{\eta,c}(t)^+ &= \left[N_{\mathcal{S}}(t)d^+\left(\hat{\mu}_{\mathcal{S}}(t), \mu_{\mathcal{S}}^{\min}\right) - c(1+\eta)\ln\left(1 + \ln N_{\mathcal{S}}(t)\right)\right] \\ X_{\eta,c}(t)^- &= \left[N_{\mathcal{S}}(t)d^-\left(\hat{\mu}_{\mathcal{S}}(t), \mu_{\mathcal{S}}^{\max}\right) - c(1+\eta)\ln\left(1 + \ln N_{\mathcal{S}}(t)\right)\right] \end{aligned}$$

Throughout the proof we use the notation $X_{\eta,c}(t)$ to refer to either $X_{\eta,c}(t)^+$ or $X_{\eta,c}(t)^-$. The cornerstone of the proof is the following Lemma 17, that tells us that $e^{\lambda X_{\eta,c}(t)}$ can be "almost" upper-bounded by some martingale.

**Lemma 17.** *Assume $1 + \eta \le e$. Fix $X_{\eta,c}(t) = X_{\eta,c}(t)^+$ or $X_{\eta,c}(t)^-$. For every $\lambda \in [0, (1+\eta)^{-1}[$ there exists a martingale $M_t^\lambda$ such that $\mathbb{E}[M_t^\lambda] = 1$ and*

$$\forall t \in \mathbb{N}^*, M_t^\lambda \ge e^{\lambda X_{\eta,c}(t) - g_{\eta,c}(\lambda)}, \tag{12}$$

*with $g_{\eta,c}(\lambda) = \lambda(1 + \eta)\ln\left(\frac{\zeta(c)}{\ln(1+\eta)^c}\right) - \ln(1 - \lambda(1 + \eta))$.*

The deviation inequality follows by combining Chernoff's method with Doob's inequality, and then carefully picking $\eta$ and $c$. For any $\lambda \in [0, (1+\eta)^{-1}[$, by Lemma 17,

$$\begin{aligned} \mathbb{P}\left(\exists t \in \mathbb{N}^* : X_{\eta,c}(t) > u\right) &\le \mathbb{P}\left(\exists t \in \mathbb{N}^* : e^{\lambda X_{\eta,c}(t)} > e^{\lambda u}\right) \\ &\le \mathbb{P}\left(\exists t \in \mathbb{N}^* : M_t^\lambda > e^{\lambda u - g_{\eta,c}(\lambda)}\right) \\ &\le \exp\left(-\left[\lambda u - g_{\eta,c}(\lambda)\right]\right). \end{aligned}$$

Then we want to apply this inequality to the best possible $\lambda$. Defining

$$g^*_{\eta,c}(u) = \max_{\lambda \in \left(0, \frac{1}{1+\eta}\right)} \left[\lambda u - g_{\eta,c}(\lambda)\right],$$

a direct computation of this Fenchel conjugate (see Lemma 20 in Appendix F.3) yields

$$g^*_{\eta,c}(u) = h\left(\frac{u}{1+\eta} - \ln\left(\frac{\zeta(c)}{(\ln(1+\eta))^c}\right)\right) - 1,$$

for $\frac{u}{1+\eta} - \ln\left(\frac{\zeta(c)}{(\ln(1+\eta))^c}\right) > 1$, where we recall that $h(x) = x - \ln(x)$.

Using the inequality[3] $\ln(1+\eta))^{-1} \le 1 + \eta^{-1}$ implies that, for $\frac{u}{1+\eta} - \ln(\zeta(c)) - c\ln\left(1 + \frac{1}{\eta}\right) \ge 1$,

$$\mathbb{P}\left(\exists t \in \mathbb{N}^* : X_{\eta,c}(t) > u\right) \le \exp\left(-\left[h\left(\frac{u}{1+\eta} - \ln(\zeta(c)) - c\ln\left(1 + \frac{1}{\eta}\right)\right) - 1\right]\right).$$

For the sake of clarity, we now pick $X_{\eta,c}(t) = X^+_{\eta,c}(t)$. Picking $\eta^* = \frac{c}{u-c}$ (that minimizes the right hand side) it holds that

$$\mathbb{P}\left(\exists t \in \mathbb{N}^* : \left[N_{\mathcal{S}}(t)d^+\left(\hat{\mu}_{\mathcal{S}}(t), \mu^{\min}_{\mathcal{S}}\right) - \frac{cu}{u-c}\ln\left(1 + \ln N_{\mathcal{S}}(t)\right)\right]^+ \ge u\right)$$

$$\le \exp\left(-\left[h\left(u - c - \ln(\zeta(c)) - c\ln\left(\frac{u}{c}\right)\right) - 1\right]\right)$$

$$\le \exp\left(-\left[h\left(ch\left(\frac{u}{c}\right) - c - \ln(\zeta(c))\right) - 1\right]\right)$$

whenever $u$ is such that $h\left(\frac{u}{c}\right) \ge 1 + \frac{1+\ln\zeta(c)}{c}$ and $1 + \eta^* = \frac{u}{u-c} \le e$.

Picking $c = 2$, for all $u \ge 6$ the three conditions $\frac{cu}{u-c} \le 3$, $h\left(\frac{u}{c}\right) \ge 1 + \frac{1+\ln\zeta(c)}{c}$ and $\frac{u}{u-c} \le e$ are satisfied and one has

$$\mathbb{P}\left(\exists t \in \mathbb{N}^* : \left[N_{\mathcal{S}}(t)d^+\left(\hat{\mu}_{\mathcal{S}}(t), \mu^{\min}_{\mathcal{S}}\right) - 3\ln\left(1 + \ln N_{\mathcal{S}}(t)\right)\right]^+ \ge u\right) \le e^{-\left[h\left(2h\left(\frac{u}{2}\right) - 2 - \ln(\zeta(2))\right) - 1\right]}$$

Picking $u$ (large enough) such that

$$h\left(2h\left(\frac{u}{2}\right) - 2 - \ln(\zeta(2))\right) - 1 = x \iff u = T(x)$$

yields inequality (6) in Theorem 7, whenever $T(x) \ge 6$. It can be checked numerically this holds for $x \ge 0.04$. Inequality (7) can be obtained following the same lines by choosing $X_{\eta,c}(t) = X^-_{\eta,c}(t)$. $\quad\square$

Proofs of intermediate results are now given in separate sections.

### F.1 Building the martingale: proof of Lemma 17

Our goal is to propose a martingale $M^\lambda_t$ that satisfies the assumptions of Lemma 17. Let

$$\phi_\mu(\lambda) = \ln \mathbb{E}_\mu[e^{\lambda X}] = b(\dot{b}^{-1}(\mu) + \lambda) - b(\dot{b}^{-1}(\mu)) \tag{13}$$

denote the cumulant generating function of the distribution that has mean $\mu$. First, it can be checked that for all $\lambda$ for which $\phi_\mu(\lambda)$ is defined, and for all arm $a$,

$$\exp\left(\lambda S_a(t) - N_a(t)\phi_{\mu_a}(\lambda)\right) \qquad \text{where} \qquad S_a(t) = \sum_{s=1}^{t} \mathbf{1}\{A_s = a\} X_s$$

is a martingale. Due to the fact that only one arm is drawn at each round, the product of these martingales for all the arms in the subset $\mathcal{S}$ is still a martingale, that can be rewritten

$$W^\lambda_t = \exp\left(\sum_{a \in \mathcal{S}} \left[S_a(t)\lambda - \phi_{\mu_a}(\lambda)N_a(t)\right]\right).$$

Moreover, $\mathbb{E}[W^\lambda_t] = 1$. We first prove the following result, that relates $X_{\eta,c}(t)$ exceeding a threshold to some $W^\lambda_t$ martingale exceeding some other threshold, for a well-chosen $\lambda$.

**Lemma 18.** *Let $i \in \mathbb{N}^*$ and $x > 0$. There exists $\lambda_i^+ = \lambda_i^+(x) < 0$ such that if $N_{\mathcal{S}}(t) \in [(1+\eta)^{i-1}, (1+\eta)^i]$ then*

$$\left\{ N_{\mathcal{S}}(t) d^+ \left( \hat{\mu}_{\mathcal{S}}(t), \mu_{\mathcal{S}}^{\min} \right) \geq x \right\} \subseteq \left\{ W_t^{\lambda_i^+} \geq e^{\frac{x}{1+\eta}} \right\}.$$

*Moreover, there exists $\lambda_i^- = \lambda_i^-(x) > 0$ such that if $N_{\mathcal{S}}(t) \in [(1+\eta)^{i-1}, (1+\eta)^i]$ then*

$$\left\{ N_{\mathcal{S}}(t) d^- \left( \hat{\mu}_{\mathcal{S}}(t), \mu_{\mathcal{S}}^{\max} \right) \geq x \right\} \subseteq \left\{ W_t^{\lambda_i^-} \geq e^{\frac{x}{1+\eta}} \right\}.$$

Lemma 18 shows that the event of interest is related to a martingale exceeding a threshold for $t$ that belongs to some *slice* $N_{\mathcal{S}}(t) \in [(1+\eta)^{i-1}, (1+\eta)^i]$. We now prove that for all $x > 0$ and $1 + \eta \leq e$, there exists a martingale $\tilde{W}_t^x$ such that $\mathbb{E}[\tilde{W}_t^x] = 1$ and

$$\left\{ X_{\eta,c}(t) - (1+\eta) \ln \left( \frac{\zeta(c)}{(\ln(1+\eta))^c} \right) \geq x \right\} \subseteq \left\{ \tilde{W}_t^x \geq e^{\frac{x}{1+\eta}} \right\}. \tag{14}$$

This martingale is one of the following *mixture martingales*:

$$\tilde{W}_t^{+,x} = \sum_{i=1}^{\infty} \gamma_i W_t^{\lambda_i^+(x+(1+\eta)\ln(1/\gamma_i))} \quad \text{and} \quad \tilde{W}_t^{-,x} = \sum_{i=1}^{\infty} \gamma_i W_t^{\lambda_i^-(x+(1+\eta)\ln(1/\gamma_i))},$$

where $\gamma_i = \frac{1}{\zeta(c)i^c}$ and $\lambda_i^\pm(x)$ are defined in Lemma 18. As $\sum_{i=1}^{\infty} \gamma_i = 1$, $\tilde{W}_t^{\pm,x}$ are martingales that satisfy $\mathbb{E}[\tilde{W}_t^{\pm,x}] = 1$. We first prove that

$$\left\{ N_{\mathcal{S}}(t) d^+ \left( \hat{\mu}_{\mathcal{S}}(t), \mu_{\mathcal{S}}^{\min} \right) - c(1+\eta)\ln(1 + \ln N_{\mathcal{S}}(t)) - (1+\eta)\ln \left( \frac{\zeta(c)}{(\ln(1+\eta))^c} \right) \geq x \right\}$$
$$\subseteq \left\{ \tilde{W}_t^{+,x} \geq e^{\frac{x}{1+\eta}} \right\}.$$

If $N_{\mathcal{S}}(t) \in [(1+\eta)^{i-1}, (1+\eta)^i]$, one can observe that $\frac{\ln N_{\mathcal{S}}(t)}{\ln(1+\eta)} \geq i - 1$, thus, for $1 + \eta \leq e$,

$$c(1+\eta)\ln(1 + \ln N_{\mathcal{S}}(t)) + (1+\eta)\ln \left( \frac{\zeta(c)}{(\ln(1+\eta))^c} \right) = (1+\eta)\ln \left( \frac{\zeta(c)(1 + N_{\mathcal{S}}(t))^c}{(\ln(1+\eta))^c} \right)$$
$$\geq (1+\eta)\ln \left( \frac{\zeta(c)(\ln(1+\eta) + N_{\mathcal{S}}(t))^c}{(\ln(1+\eta))^c} \right) = (1+\eta)\ln \left( \zeta(c)\left[ 1 + \frac{\ln N_{\mathcal{S}}(t)}{(\ln(1+\eta))} \right]^c \right)$$
$$\geq (1+\eta)\ln \left( \zeta(c)i^c \right) = (1+\eta)\ln \frac{1}{\gamma_i}$$

Thus for $N_{\mathcal{S}}(t) \in [(1+\eta)^{i-1}, (1+\eta)^i]$, it holds using Lemma 18 that

$$\left\{ N_{\mathcal{S}}(t) d^+ \left( \hat{\mu}_{\mathcal{S}}(t), \mu_{\mathcal{S}}^{\min} \right) - c(1+\eta)\ln(1 + \ln N_{\mathcal{S}}(t)) - (1+\eta)\ln \left( \frac{\zeta(c)}{(\ln(1+\eta))^c} \right) \geq x \right\}$$
$$\subseteq \left\{ N_{\mathcal{S}}(t) d^+ \left( \hat{\mu}_{\mathcal{S}}(t), \mu_{\mathcal{S}}^{\min} \right) \geq x + (1+\eta)\ln \frac{1}{\gamma_i} \right\}$$
$$\subseteq \left\{ \tilde{W}_t^{\lambda_i^+(x+(1+\eta)\ln \frac{1}{\gamma_i})} \geq e^{\frac{x}{1+\eta} + \ln(1/\gamma_i)} \right\} = \left\{ \gamma_i \tilde{W}_t^{\lambda_i^+(x+(1+\eta)\ln \frac{1}{\gamma_i})} \geq e^{\frac{x}{1+\eta}} \right\}$$
$$\subseteq \left\{ \tilde{W}_t^{+,x} \geq e^{\frac{x}{1+\eta}} \right\}.$$

Similarly, one can prove that

$$\left\{ N_{\mathcal{S}}(t) d^- \left( \hat{\mu}_{\mathcal{S}}(t), \mu_{\mathcal{S}}^{\max} \right) - c(1+\eta)\ln(1 + \ln N_{\mathcal{S}}(t)) - (1+\eta)\ln \left( \frac{\zeta(c)}{(\ln(1+\eta))^c} \right) \geq x \right\}$$
$$\subseteq \left\{ \tilde{W}_t^{-,x} \geq e^{\frac{x}{1+\eta}} \right\}.$$

Let $C(\eta) = \frac{\zeta(c)}{(\ln(1+\eta))^c}$. For all $\lambda > 0$ and $z > 1$ it follows from inequality (14) that

$$\left\{ e^{\lambda(X_{\eta,c}(t) - (1+\eta)\ln C(\eta))} \geq z \right\} \subseteq \left\{ \tilde{W}^{\frac{1}{\lambda}\ln(z)} \geq e^{\frac{\ln(z)}{\lambda(1+\eta)}} \right\} = \left\{ e^{-\frac{\ln(z)}{\lambda(1+\eta)}} \tilde{W}^{\frac{1}{\lambda}\ln(z)} \geq 1 \right\}$$

Letting $\overline{W}_t^{\lambda,z} = e^{-\frac{\ln(z)}{\lambda(1+\eta)}} \tilde{W}^{\frac{1}{\lambda}\ln(z)}$, $\overline{W}_t^{\lambda,z}$ is a martingale that satisfies $\mathbb{E}\left[\overline{W}_t^{\lambda,z}\right] = e^{-\frac{\ln(z)}{\lambda(1+\eta)}}$. For all $\lambda \in [0, 1/(1+\eta)[$, we now define

$$\overline{M}_t^\lambda = 1 + \int_1^\infty \overline{W}_t^{\lambda,z} dz$$

Using that $\overline{W}_t^{\lambda,z} \geq \mathbb{1}_{\left(e^{\lambda(X_{\eta,c}(t)-(1+\eta)\ln C(\eta))}\geq z\right)}$ and the expression of $\mathbb{E}\left[\overline{W}_t^{\lambda,z}\right]$ yields

$$\overline{M}_t^\lambda \geq e^{\lambda(X_{\eta,c}(t)-(1+\eta)\ln C(\eta))} \quad \text{and} \quad \mathbb{E}\left[\overline{M}_t^\lambda\right] = \frac{1}{1-\lambda(1+\eta)}.$$

From there, we obtain that $M_t^\lambda = (1-\lambda(1+\eta))\overline{M}_t^\lambda$ satisfies $\mathbb{E}[M_t^\lambda] = 1$ and

$$M_t^\lambda \geq e^{\lambda X_{\eta,c}(t)-\lambda(1+\eta)\ln C(\eta)+\ln(1-\lambda(1+\eta))},$$

which concludes the proof.

## F.2 Proof of Lemma 18

Let $\nu_{\min}$ be the natural parameter such that $\mu_{\mathcal{S}}^{\min} = \dot{b}(\nu_{\min})$ and $\nu_{\max}$ be the natural parameter such that $\mu_{\mathcal{S}}^{\max} = \dot{b}(\nu_{\max})$. Define $\lambda_i^- > 0$ and $\lambda_i^+ < 0$ such that

$$\text{KL}(\nu_{\min} + \lambda_i^+, \nu_{\min}) = \frac{x}{(1+\eta)^i} \quad \text{and} \quad \text{KL}(\nu_{\max} + \lambda_i^-, \nu_{\max}) = \frac{x}{(1+\eta)^i},$$

where $\text{KL}(\nu, \nu')$ is the Kullback-Leibler divergence between the distributions of natural parameter $\nu$ and $\nu'$. Defining $\mu_i^+ := \dot{b}^{-1}(\nu + \lambda_i^+) < \mu_{\mathcal{S}}^{\min}$ and $\mu_i^- := \dot{b}^{-1}(\nu + \lambda_i^-) > \mu_{\mathcal{S}}^{\max}$ and using some properties of the KL-divergence for exponential families, one can write

$$
\begin{aligned}
d(\mu_i^+, \mu_{\mathcal{S}}^{\min}) &= \text{KL}(\nu_{\min} + \lambda_i^+, \nu_{\min}) = \lambda_i^+ \mu_i^+ - \phi_{\mu_{\mathcal{S}}^{\min}}(\lambda_i^+) \\
d(\mu_i^-, \mu_{\mathcal{S}}^{\max}) &= \text{KL}(\nu_{\max} + \lambda_i^-, \nu_{\max}) = \lambda_i^- \mu_i^- - \phi_{\mu_{\mathcal{S}}^{\max}}(\lambda_i^-).
\end{aligned}
$$

For $N_{\mathcal{S}}(t) \in [(1+\eta)^{i-1}, (1+\eta)^i]$, one can write (using notably that $\lambda_i^+$ is negative)

$$
\begin{aligned}
\left\{N_{\mathcal{S}}(t)d^+(\hat{\mu}_{\mathcal{S}}(t), \mu_{\mathcal{S}}^{\min}) \geq x\right\} &\subseteq \left\{d^+(\hat{\mu}_{\mathcal{S}}(t), \mu_{\mathcal{S}}^{\min}) \geq \frac{x}{(1+\eta)^i}\right\} \\
&\subseteq \{\hat{\mu}_{\mathcal{S}}(t) \leq \mu_i^+\} \\
&\subseteq \left\{\lambda_i^+ \hat{\mu}_{\mathcal{S}}(t) - \phi_{\mu_{\mathcal{S}}^{\min}}(\lambda_i^+) \geq \lambda_i^+ \mu_i^+ - \phi_{\mu_{\mathcal{S}}^{\min}}(\lambda_i^+) = \text{KL}(\nu_{\min} + \lambda_i^+, \nu_{\min})\right\} \\
&\subseteq \left\{\lambda_i^+ \hat{\mu}_{\mathcal{S}}(t) - \phi_{\mu_{\mathcal{S}}^{\min}}(\lambda_i^+) \geq \frac{x}{(1+\eta)^i}\right\} \\
&\subseteq \left\{\lambda_i^+ N_{\mathcal{S}}(t)\hat{\mu}_{\mathcal{S}}(t) - N_{\mathcal{S}}(t)\phi_{\mu_{\mathcal{S}}^{\min}}(\lambda_i^+) \geq \frac{x}{1+\eta}\right\} \\
&\subseteq \left\{\lambda_i^+ \sum_{a\in\mathcal{S}} S_a(t) - \left(\sum_{a\in\mathcal{S}} N_a(t)\right)\phi_{\mu_{\mathcal{S}}^{\min}}(\lambda_i^+) \geq \frac{x}{1+\eta}\right\}
\end{aligned}
$$

Now using Lemma 19 below, that can easily be checked by differentiating the equality (13), one can use that as $\lambda_i^+ < 0$,

$$\forall a \in \mathcal{S}, \phi_{\mu_{\mathcal{S}}^{\min}}(\lambda_i^+) \geq \phi_{\mu_a}(\lambda_i^+).$$

Therefore, it follows that

$$\left\{N_{\mathcal{S}}(t)d^+(\hat{\mu}_{\mathcal{S}}(t), \mu_{\mathcal{S}}^{\min}) \geq x\right\} \subseteq \left\{\lambda_i^+ \sum_{a\in\mathcal{S}} S_a(t) - \sum_{a\in\mathcal{S}} N_a(t)\phi_{\mu_a}(\lambda_i^+) \geq \frac{x}{1+\eta}\right\} = \left\{W_t^{\lambda_i^+} \geq \frac{x}{1+\eta}\right\}$$

This proves the first inclusion in Lemma 18. The proof of the second inclusion follows exactly the same lines, using this time that $\lambda_i^- > 0$:

$$
\begin{aligned}
\{N_{\mathcal{S}}(t)d^-(\hat{\mu}_{\mathcal{S}}(t), \mu_{\mathcal{S}}^{\max}) \geq x\} \quad &\subseteq \quad \{\hat{\mu}_{\mathcal{S}}(t) \geq \mu_i^-\} \\
&\subseteq \quad \left\{\lambda_i^- \hat{\mu}_{\mathcal{S}}(t) - \phi_{\mu_{\mathcal{S}}^{\max}}(\lambda_i^-) \geq \frac{x}{(1+\eta)^i}\right\} \\
&\subseteq \quad \left\{\lambda_i^- \sum_{a \in \mathcal{S}} S_a(t) - \left(\sum_{a \in \mathcal{S}} N_a(t)\right) \phi_{\mu_{\mathcal{S}}^{\max}}(\lambda_i^-) \geq \frac{x}{1+\eta}\right\} \\
&\subseteq \quad \left\{\lambda_i^- \sum_{a \in \mathcal{S}} S_a(t) - \sum_{a \in \mathcal{S}} N_a(t)\phi_{\mu_a}(\lambda_i^-) \geq \frac{x}{1+\eta}\right\},
\end{aligned}
$$

where the last inequality uses that by Lemma 19 $\forall a \in \mathcal{S}, \phi_{\mu_{\mathcal{S}}^{\max}}(\lambda_i^-) \geq \phi_{\mu_a}(\lambda_i^-)$.

**Lemma 19.** *The mapping $\mu \mapsto \phi_\mu(\lambda)$ is non-increasing if $\lambda < 0$ and non-decreasing if $\lambda > 0$.*

### F.3 Technical Results

**Lemma 20.** *Define $g(\lambda) = A\lambda - \ln(1 - \lambda B)$ for $\lambda \in [0, B^{-1}[$. Then if $\frac{u-A}{B} \geq 1$,*

$$
g^*(u) = \max_{\lambda \in [0, B^{-1}[} [\lambda u - g(\lambda)] = h\left(\frac{u-A}{B}\right) - 1,
$$

*where $h(u) = u - \ln u$.*

*Proof.* Differentiating shows that $\lambda \mapsto \lambda u - g(\lambda)$ attains its maximum on in $\lambda^* = \frac{1}{B} - \frac{1}{x-A}$, which also satisfies $1 - \lambda^* B = \frac{B}{x-A}$. If $\frac{u-A}{B} \geq 1$, $\lambda^* \in [0, B^{-1}[$ and one obtains

$$
\begin{aligned}
g^*(u) &= \lambda^* u - g(\lambda^*) = \lambda^*(u - A) + \ln(1 - \lambda^* B) \\
&= \frac{x-A}{B} - \ln \frac{x-A}{B} - 1 \\
&= h\left(\frac{x-A}{B}\right) - 1.
\end{aligned}
$$

$\square$

The next result permits to derive a tight upper bound on the threshold function $T$ featured in Theorem 7. Recall this function is defined in terms of the inverse mapping of $h : [1, +\infty[ \to \mathbb{R}^*$ defined by $h(u) = u - \ln(u)$.

**Lemma 21.** *$h$ is increasing on $[1, +\infty[$ and its inverse function, defined on $[1, +\infty[$ can be expressed in terms of negative branch of the Lambert function: $h^{-1}(x) = -W_{-1}(-e^{-x})$. The following inequality holds:*

$$
\forall x \geq 1, \quad h^{-1}(x) \leq x + \ln(x + \sqrt{2(x-1)}).
$$

*Proof.* We may write

$$
h^{-1}(x) = \inf_{z \geq 1} z\left(x - 1 + \ln \frac{z}{z-1}\right)
$$

Plugging in the sub-optimal feasible choice $z = 1 + \frac{1}{(x-1)+\sqrt{2(x-1)}}$ reveals

$$
\begin{aligned}
h^{-1}(x) &\leq \left(1 + \frac{1}{(x-1) + \sqrt{2(x-1)}}\right)\left(x - 1 + \ln\left(x + \sqrt{2(x-1)}\right)\right) \\
&\leq 1 + (x-1) + \ln\left(x + \sqrt{2(x-1)}\right).
\end{aligned}
$$

Where the last inequality uses $\ln\left(x + \sqrt{2(x-1)}\right) \leq \sqrt{2(x-1)}$ which holds with equality at $x = 1$ and whose gap is increasing (as can be checked by differentiation). $\square$

# G Aggregate Stopping Rule is $\delta$-correct (Lemma 8)

First assume $\boldsymbol{\mu} \in \mathcal{H}_>$. Then the probability of error is upper bounded by

$$
\begin{aligned}
&\mathbb{P}\left(\exists t \in \mathbb{N}, \exists \mathcal{S} : N_{\mathcal{S}}(t) d^+\left(\hat{\mu}_{\mathcal{S}}(t), \theta\right) \geq 3\ln(1 + \ln(N_{\mathcal{S}}(t))) + T\left(\ln(1/(\delta\pi(\mathcal{S})))\right)\right) \\
\leq\ &\sum_{\mathcal{S}} \mathbb{P}\left(\exists t \in \mathbb{N} : N_{\mathcal{S}}(t) d^+\left(\hat{\mu}_{\mathcal{S}}(t), \theta\right) \geq 3\ln(1 + \ln(N_{\mathcal{S}}(t))) + T\left(\ln(1/(\delta\pi(\mathcal{S})))\right)\right) \\
\leq\ &\sum_{\mathcal{S}} \mathbb{P}\left(\exists t \in \mathbb{N} : N_{\mathcal{S}}(t) d^+\left(\hat{\mu}_{\mathcal{S}}(t), \min_{a\in\mathcal{S}} \mu_a\right) \geq 3\ln(1 + \ln(N_{\mathcal{S}}(t))) + T\left(\ln(1/(\delta\pi(\mathcal{S})))\right)\right) \\
\leq\ &\sum_{\mathcal{S}} \delta\pi(\mathcal{S}) = \delta.
\end{aligned}
$$

The second inequality uses that on $\mathcal{H}_<$, all $\mu_a$ are larger than $\gamma$ and $x \mapsto d^+\left(\hat{\mu}_{\mathcal{S}}(t), x\right)$ is non-decreasing. The last inequality follows from the first inequality in Theorem 7.

Now assume $\boldsymbol{\mu} \in \mathcal{H}_<$: there exists $a$ such that $\mu_a < \gamma$. The probability of error is upper bounded by

$$
\begin{aligned}
&\mathbb{P}\left(\exists t \in \mathbb{N}, \forall a,\ N_a(t) d^-\left(\hat{\mu}_a(t), \gamma\right) \geq 3\ln(1 + \ln(N_a(t))) + T\left(\ln(1/\delta)\right)\right) \\
\leq\ &\mathbb{P}\left(\exists t \in \mathbb{N} : N_a(t) d^-\left(\hat{\mu}_a(t), \gamma\right) \geq 3\ln(1 + \ln(N_a(t))) + T\left(\ln(1/\delta)\right)\right) \\
\leq\ &\mathbb{P}\left(\exists t \in \mathbb{N} : N_a(t) d^-\left(\hat{\mu}_a(t), \mu_a\right) \geq 3\ln(1 + \ln(N_a(t))) + T\left(\ln(1/\delta)\right)\right) \leq \delta.
\end{aligned}
$$

The second inequality holds as $\mu_a < \gamma$ and $x \mapsto d^-\left(\hat{\mu}_a(t), \gamma\right)$ is non-increasing. The last inequality is an application of the second inequality of Theorem 7, for singleton $\mathcal{S} = \{a\}$.