[Reviews · NeurIPS 2018]

Reviewer 1



This paper considers a variant of the pure exploration multi-armed bandit problem, where we are given a threshold gamma and we wish to identify whether the minimum mean among the arms is larger or smaller than gamma. An algorithm is designed that identifies the correct hypothesis with probability at least 1-delta. Lower bounds on expected sample size are also provided, and the algorithm's expected sample size is shown to match (exactly, not up to constants) asymptotically in the limit as delta goes to 0. The paper assumes that the sampling distributions are independent and from a known exponential family. I'm positive on this paper. It contains a thorough analysis and a novel algorithm with strong results for this problem. One could object to the strength of the parametric sampling assumption, but this is offset by the precision of the analysis, and I find the assumptions to be fine. There is an older literature on something called "multiple comparisons with a standard" and "multiple comparisons with the best". In this problem, we have a collection of arms, typically with independent sampling distributions in exponential families, and we wish to make statistically valid statements about the difference between each arm's mean and either some known value (gamma) or the best arm. In particular, here is a quote from the abstract of Kim and Nelson (2005): "The goal is to find systems whose expected performance measures are larger or smaller than a single system referred to as a standard and, if there is any, to find the one with the largest or smallest performance. Our procedures allow.. known or unknown expected performance of the standard." So it seems that this literature is very close to the problem considered here, but not identical. While Kim and Nelson seem to do hypothesis testing of mu^* > gamma, the paper seems different because their procedure does additional statistical inference. The paper would better serve to teach the ML community about older relevant literature if it reviewed the most relevant of this literature. A survey into the literature is Goldsman and Nelson (1998). Goldsman, D., and B. L. Nelson. 1998. Comparing Systems via Simulation. In Handbook of simulation: Principles, Methodology, Advances, Applications, and Practice, ed. J. Banks, 273–306. New York: John Wiley & Sons. Kim, Seong-Hee. "Comparison with a standard via fully sequential procedures." ACM Transactions on Modeling and Computer Simulation (TOMACS) 15.2 (2005): 155-174. Another criticism is that the problem considered seems unusual. Are there applicatios where I would want to know whether mu* > gamma, but I wouldn't also want to know the identity of the best arm? Overall, I found the paper very clear, and the analysis novel and of high quality. Significance is affected by the applicability of the problem considered. Here are a few typos and small comments: - Are conditions needed on the prior? I don't see any stated, but at the bare minimum it seems that we need the prior to have support on both H_> and H_<, e.g., on lines 201-202 in the proof of Theorem 5. - In equation (1), I believe delta is not defined. I believe it is the simplex. - In prop 2, i don't think N_a(tau) has been defined. - theorem 3 uses d_+ but this hasn't been defined yet. I assume this is d^+ defined in the next section. - line 258 "permits to build" should be "permits building" - there is something wrong with reference [3] ##### UPDATE AFTER READING AUTHOR RESPONSE #### My most substantive concern on this paper was the strength of its motivating application. To be clear, I don't consider a clear demonstration of applicability to be a necessary condition for acceptance at NIPS, so I see this as optional. Indeed,by far the primary contribution of this paper is theoretical. After reading the author response, I re-read the relevant paragraphs about applicability to Monte Carlo Tree Search and applications in quality control on pages 1-2. I am somewhat more positive now than I was at the original writing of my review about applicability, though I think there are still many unanswered questions. Indeed, to truly be convinced I would want to see experiments using real data and more discussion of the application context. Here are some questions that I have relevant to my original comment quoted here: "Another criticism is that the problem considered seems unusual. Are there applicatios where I would want to know whether mu* > gamma, but I wouldn't also want to know the identity of the best arm?" In the Monte Carlo Tree Search application, the paper unfortunately doesn't provide enough details given my background to evaluate whether the problem formulation really does have direct applicability to MCTS. I agree that in spirit it is close, but after skimming the given reference [33] I wonder: - in 33, pruning (Algorithm 3) is done by comparing the estimated min/max value of a particular child node against the root. There is error in estimating the min/max value of the root while applying the method in this paper would treat it as a constant (gamma). - in 33, samples of a child nodes value don't seem to come iid, but instead are biased through hierarchical estimation: the value of node u is estimated as the min/max value of its child nodes. Even if the estimates of the values of the leaf nodes are unbiased, it seems that estimates will be biased higher in the tree. - in MCTS, if I sample 1 child node u, then that can improve the estimates for another node v at the same level because it can help estimate the value of nodes at lower levels shared by these nodes u and v. - is the measure of correctness used really the most appropriate thing for MCTS? The number of nodes tested is extremely large in MCTS, and it seems that the sample complexity required to make every decision correctly with reasonable probability would just be too large. Instead, it seems like one would expect some errors, and one would wish that the size of the errors wouldn't be too large most of the time. This would seem to suggest an error measure that put more weight on incorrect conclusions for arms with large gaps between the value and gamma. In the quality control applications, there is also room for improving the clarity of the application context. For example: - regarding the e-learning application, we typically grade students by estimating their average performance across a range of subjects rather than their minimum. Are there existing e-learning settings where performance is certified in this way? - when certifying a batch of machines has a minimum probability of successfully producing a widget, I'd feel better if I had more detail about a situation where one would want to do this. Does this arise for regulatory reasons? If a plant owner were doing this, and found that the minimum probability fell below a threshold, then she would probably want to know which machine was causing the problem so that she could fix / replace it. If it is for regulatory reasons, why do they care about the worst machine instead of caring about the fraction of the widgets (produced by all the machines) that have quality below a threshold? Is the proposed loss function the best one? - unfortunately I do not understand the anomaly detection setting on line 38. - In crowdsourcing, when establishing the quality of workers, what would I do if I found out that a cohort contained a careless one? Would I simply go get a new cohort? Would I have some other non-crowd workers do the task I originally planned for the crowd? It seems like an approach that is more efficient than each of these would be to try to figure out which one(s) were careless and ignore their work. At the same time, I feel that the motivating quality control applications are nearly on par in terms of actual applicability with many other theoretical bandit papers published in the past at NIPS. (For example, the assumption of each arm having iid rewards common in the stochastic bandits literature is routinely violated in applications.) Finally, the Bernoulli argument is a good one in favor of the parametric assumption.

Reviewer 2



Summary of the paper: The paper considers a sequential hypothesis testing of the following form. Given a set of arms with unknown mean reward, the problem is to identify whether the minimum mean reward is below a given threshold or above it. The goal is to develop an algorithm which can issue a high probability correct response with minimum number of arm samplings. The paper first establishes a lower bound on the sample complexity of this problem. Then, a new sampling algorithm (Murphy Sampling) is proposed as the solution. It has been proved that the proposed sampling algorithm has a runtime matching the lower bound. The performance of the proposed algorithm is validated through simulation study. My Assessment: The paper has been very well written with high clarity. All the assumptions and details are concretely and clearly stated. The paper’s approach of developing the algorithm is to start with the lower bound on the sample complexity of the problem. Based on this lower bound, two already known algorithms are shown to achieve the optimal sample complexity, each only in one of the two possible hypotheses. Then, a modified version of Thompson sampling (called Murphy sampling) is shown to work almost the same in both scenarios. I think originality of this idea is above average. Also, the significance and quality of the paper is also above average. In order to establish efficient stopping rules, the paper introduces new concentration inequalities which can be of independent interest. One very important issue is in using rejecting sampling when implementing Murphy sampling. As has been noted by the authors, the rejection probability could cause a problem in this implementation. Although it has not caused any problem in the experiments run for the paper, but there could be scenarios where the rejection probability prohibitively large. >>> Update after reading author's response: I think the authors have provided a convincible response to my question. I am changing my score to 7. As in my first review, I think this paper is theoretically very strong and the analysis is concrete with novelties. However, the problem is not motivated as strongly. I think adding more discussions on the possible applications would make the paper even more appealing.

Reviewer 3



This submission tries to study the sequential adaptive hypothesis test for the lowest mean among a finite set of distribution to propose Murphy sampling, compared with LCB and TS with different stopping rules Agg, Box, GLRT, the experiments were conducted in a really small scale and on the artificial data only. Remarks: 1. consider this is a purely theoretical work, it's unclear what's the new proof techniques or strategies you contributed rather than the similar martingale ones 2. do not point to any figures that are not in the main text, put the key things you want to show in the main body. 3. section 7 is not really a discussion, and the conclusion is missing, it also reflects this work is inconclusive. 4. the authors are unable to provide any real experiments evidence to match the claims, you only play with tiny toy data which is clearly not enough for a high-quality ML paper. 5. line 297 you stated 'large-scale', but you merely reported results in 500 repetitions, you need to spend more time to study how to execute the useful experiments. 6. there is a huge distance between the asymptotic analysis and real applications, in order to mitigate this gap there is a significant amount of work need to be done in order to roll it on board 7. this manuscript is not well written, in the introduction, it seems more like part of an abstract, through the whole draft, you want to position and rectify the whole draft systematically 8. last but not least, there are quite a number of typos and grammar errors irritating during the read, please carefully fix them as well To conclude, this draft did not make significant contributions to this field, neither in theory nor practice. %updated comments after rebuttal They have not used the author response style file, but I read their rebuttal, and as they admitted that there can be no serious problem of computational complexity due to a too high rejection probability, which needs further evidence to support their claim. Some actions that will lead to a certain reward, this simpler case makes less sense to identify which one it is, their unnatural setting limits the applicability. Nevertheless, I believe the theoretical contribution is okay to this field, though clearly the empirical contribution has more space to improve.